# Effects of relaxation interventions during pregnancy on maternal mental health, and pregnancy and newborn outcomes: A systematic review and meta-analysis

**Mubarek Abera**[1]*, **Charlotte Hanlon**[2,3], **Beniam Daniel**[4], **Markos Tesfaye**[5,6], **Abdulhalik Workicho**[7], **Tsinuel Girma**[8], **Rasmus Wibaek**[9], **Gregers S. Andersen**[9], **Mary Fewtrell**[10], **Suzanne Filteau**[11], **Jonathan C. K. Wells**[10]

1 Department of Psychiatry, Faculty of Medical Science, Jimma University, Jimma, Ethiopia, 2 Centre for Global Mental Health, Health Service and Population Research Department, Institute of Psychiatry, Psychology and Neuroscience, King's College London, London, United Kingdom, 3 Department of Psychiatry, School of Medicine, College of Health Sciences, Addis Ababa University, Addis Ababa, Ethiopia, 4 School of Nursing, College of Medicine and Health Sciences, Arbaminch University, Arbaminch, Ethiopia, 5 NORMENT, Division of Mental Health and Addiction, Oslo University Hospital & Institute of Clinical Medicine, University of Oslo, Oslo, Norway, 6 NORMENT, Department of Clinical Science, University of Bergen, Bergen, Norway, 7 Department of Epidemiology, Faculty of Public Health, Jimma University, Jimma, Ethiopia, 8 Department of Pediatrics, Faculty of Medical Sciences, Jimma University, Jimma, Ethiopia, 9 Clinical Epidemiology Research, Steno Diabetes Center Copenhagen, Herlev, Denmark, 10 Population, Policy and Practice Research and Teaching Department, UCL Great Ormond Street Institute of Child Health, London, United Kingdom, 11 Faculty of Epidemiology and Population Health, London School of Hygiene and Tropical Medicine, London, United Kingdom

* mubarek.abera@ju.edu.et, abmubarek@gmail.com

**Data Availability Statement:** All relevant data are within the manuscript and its Supporting Information files.

## Abstract

### Background

Stress during pregnancy is detrimental to maternal health, pregnancy and birth outcomes and various preventive relaxation interventions have been developed. This systematic review and meta-analysis aimed to evaluate their effectiveness in terms of maternal mental health, pregnancy and birth outcomes.

### Method

The protocol for this review is published on PROSPERO with registration number CRD42020187443. A systematic search of major databases was conducted. Primary outcomes were maternal mental health problems (stress, anxiety, depression), and pregnancy (gestational age, labour duration, delivery mode) and birth outcomes (birth weight, Apgar score, preterm birth). Randomized controlled trials or quasi-experimental studies were eligible. Meta-analyses using a random-effects model was conducted for outcomes with sufficient data. For other outcomes a narrative review was undertaken.

### Result

We reviewed 32 studies comprising 3,979 pregnant women aged 18 to 40 years. Relaxation interventions included yoga, music, Benson relaxation, progressive muscle relaxation

**Funding:** National Institute of Health Research through the NIHR Global Health Research Group - NIHR134325 - NIHR200842 Wellcome Trust - 222154/Z20/Z - 223615/Z/21/Z - Dr Charlotte Hanlon.

**Competing interests:** The authors have declared that no competing interests exist.

(PMR), deep breathing relaxation (BR), guided imagery, mindfulness and hypnosis. Intervention duration ranged from brief experiment (~10 minutes) to 6 months of daily relaxation. Meta-analyses showed relaxation therapy reduced maternal stress (-4.1 points; 95% Confidence Interval (CI): -7.4, -0.9; 9 trials; 1113 participants), anxiety (-5.04 points; 95% CI: -8.2, -1.9; 10 trials; 1965 participants) and depressive symptoms (-2.3 points; 95% CI: -3.4, -1.3; 7 trials; 733 participants). Relaxation has also increased offspring birth weight (80 g, 95% CI: 1, 157; 8 trials; 1239 participants), explained by PMR (165g, 95% CI: 100, 231; 4 trials; 587 participants) in sub-group analysis. In five trials evaluating maternal physiological responses, relaxation therapy optimized blood pressure, heart rate and respiratory rate. Four trials showed relaxation therapy reduced duration of labour. Apgar score only improved significantly in two of six trials. One of three trials showed a significant increase in birth length, and one of three trials showed a significant increase in gestational age. Two of six trials examining delivery mode showed significantly increased spontaneous vaginal delivery and decreased instrumental delivery or cesarean section following a relaxation intervention.

## Discussion

We found consistent evidence for beneficial effects of relaxation interventions in reducing maternal stress, improving mental health, and some evidence for improved maternal physiological outcomes. In addition, we found a positive effect of relaxation interventions on birth weight and inconsistent effects on other pregnancy or birth outcomes. High quality adequately powered trials are needed to examine impacts of relaxation interventions on newborns and offspring health outcomes.

## Conclusion

In addition to benefits for mothers, relaxation interventions provided during pregnancy improved birth weight and hold some promise for improving newborn outcomes; therefore, this approach strongly merits further research.

## Introduction

Stress, defined as "a state of mental discomfort, unpleasant feeling, worry or tension", when occurring during pregnancy is a major public health problem in low- and middle-income countries (LMICs), associated with adverse maternal health, pregnancy and birth outcomes [1, 2]. Stress occurs when a demand to deal with internal or external cues/stressors exceeds the coping skills and resilience of individuals [3]. Common stressors during pregnancy include physical stressors, such as illness and discomfort, changes in lifestyle, poor social support, unplanned pregnancy, low financial income, role transitions, hormonal and physiological changes, anticipation of labour and delivery, and intimate partner violence during and after pregnancy [4, 5]. Stress can be acute, episodic/transient or chronic, depending on the type and nature of stressors [6].

The human body stores unresolved psychological stress in the musculoskeletal system, mainly in the scalp, neck, back, chest, abdomen and extremities [7]. This can result in sustained contraction of the muscles which interferes with normal physiological functions [7]. The resulting stress response in the body involves psychological (mental, emotional or behavioral) and/or physiological responses (blood pressure, heart rate, respiratory rate and body temperature) [8]. Biologically, stress activates the Hypothalamus-Pituitary-Adrenal (HPA)

axis and the immune system through which it increases circulating glucocorticoids and pro-inflammatory markers [8]. Stress-induced glucocorticoid in the brain interferes with normal neurogenesis and synaptic plasticity leading to impaired functions of the nervous system which can result in mental illness [9–11]. This is recognized as the body-mind connection [12–14] whereby the body and the mind work together to maintain optimal psychological equilibrium and physiological homeostasis.

Stress during pregnancy can negatively impact maternal health and well-being [15] and generally increases the risk of non-communicable diseases such as hypertension, diabetes, cardiovascular problems, anxiety and depression [16]. Nearly one in three women globally [17], and more than half of women in LMICs experience stress during their pregnancy [17–20]. In Ethiopia, pregnant women experience higher levels of psychological stress compared to non-pregnant women and also exhibit lower resilience [18]. Globally, 15 to 25% of women experience high levels of anxiety or depressive symptoms during pregnancy [21, 22], with higher estimates from studies conducted in LMICs [22, 23]. Stress during pregnancy can affect the maternal immune system and increase the risk of infection and inflammatory diseases leading to maternal physical ill-health during and after pregnancy [8]. Antenatal stress and maternal mental disorders can adversely affect normal growth and development of the fetus and result in unfavorable pregnancy, obstetric and birth outcomes [15, 23, 24]. It can also influence the post-natal physical, mental and neurobehavioral health of the offspring, potentially leading to an increased risk of non-communicable diseases including mental illness later in life [24].

Several intervention modalities, including psychotropic medications, relaxation therapy and psychosocial and counseling therapies have been tested to reduce stress and improve the mental health of pregnant women [25]. Treatment of anxiety or depression with psychotropic medications during pregnancy or lactation carries potential risks for the mother and her offspring and has low acceptability [25]. Thus non-pharmacological interventions, such as counseling or relaxation therapies, are preferred for stress management during pregnancy [26, 27]. However, no comprehensive review of evidence is available on the effectiveness of relaxation interventions provided during pregnancy on maternal and neonatal health outcomes. This paper therefore aimed to systematically synthesize evidence on the effects of relaxation interventions on maternal stress and mental health during pregnancy and on pregnancy and birth outcomes.

## Methods

### Protocol registration

The protocol for this review was registered at PROSPERO International prospective register of systematic reviews and can be accessed at: https://www.crd.york.ac.uk/prospero/display_record.php?ID=CRD42020187443.

### Article selection

The review process followed the Preferred Reporting Items for Systematic Review and Meta-Analysis (PRISMA) guideline [28]. To identify relevant articles, a three-step search strategy was employed. In the first step, key free text and MeSH terms were identified and developed. Then a comprehensive search was conducted in the following major databases: PubMed, EMBASE Classic + EMBASE (Ovid), MEDLINE in-process and non-indexed citations, MEDLINE daily, and MEDLINE (Ovid), Cumulative Index to Nursing & Allied Health Plus (CINAHL via EBSCO) and the Cochrane library. In addition, a manual search was conducted to identify further relevant studies from the reference lists of identified studies. Unpublished and grey literature were excluded.

The search terms were developed with a combination of key words relating to the study population, intervention types and outcome indicators, as follows. ("Pregnant women" OR "pregnancy" OR "prenatal" OR "prenatal care" OR "mother" OR "antenatal" OR "antenatal care" OR "maternal" OR "maternal care") AND ("Relaxation therapy" OR "Mindfulness therapy" OR "Progressive muscle relaxation (PMR) therapy" OR "Music therapy" OR "Exercise therapy" OR "deep breathing relaxation therapy" OR "Meditation therapy" OR "hypnosis therapy" OR "relaxation lighting"), AND ("Stress" OR "distress" OR "anxiety" OR "depression" OR "Birth-weight" OR "birth weight" OR "birth outcome" OR "Apgar", "Apgar score", "Gestation", OR "Gestational age at birth").

Studies were eligible if they employed Randomised Controlled Trial (RCT) or quasi-experimental designs, applied a relaxation intervention during pregnancy or labour, were published in English, and reported one or more of the outcomes of interest specified in our search strategy. Observational studies (case reports, cross-sectional and cohort studies) and editorials or opinion pieces were excluded.

**PICO.** *Population*: apparently healthy pregnant women.

*Intervention/exposure*: stress reduction relaxation therapy. Any form of relaxation intervention, whether mind-based (tapes, music, meditation) or physical/body-based (massage, stretch or exercise) including progressive muscle relaxation (PMR) and deep breathing exercises, that were applied during pregnancy with the aim of reducing stress and promoting mental health.

*Comparators/controls*: pregnant women who did not receive a stress-reduction relaxation intervention but who received treatment as usual.

*Outcomes*: the main outcomes were measures of stress (self-report, physiological or biochemical), mental health problems (anxiety or depressive symptoms), obstetrics/pregnancy outcomes (gestational age, mode of delivery, duration of labour), birth outcomes (birth weight, birth length, Apgar score) and maternal physiology (vital signs).

*Timing of outcome measures*: studies that measured the outcome during, immediately after, or some weeks or months after the intervention were included.

**Study screening process.** The literature search was concluded on 26 August 2023. To decide on inclusion, the articles were first screened by title and then by abstract using the eligibility criteria. Full texts of the selected articles were then assessed based on the inclusion and exclusion criteria. Two authors (MA and BD) screened all articles for eligibility. Any queries were discussed with one additional author (AW) to reach a consensus. The screening process and reasons for exclusion are documented.

**Methodological quality assessment.** Two independent assessors (BD and MA) evaluated the methodological quality of studies in terms of randomization, masking and availability of descriptions for withdrawal and dropout of all participants based on the modified Jadad scoring scale [29] and the modified Delphi List Criteria [30] to assess the overall quality of the studies. Using the Cochrane Collaboration's Assessment checklist [31], the risk of bias was assessed and rated as low, high or unclear for individual elements relating to five domains (selection, performance, attrition, reporting and other). The criterion on blinding was excluded as it is usually impossible to conduct relaxation therapy while blinding the participant or the care providers. S1 Table shows risk of bias assessment for all included studies.

## Data extraction

The findings were extracted using a standard data extraction form prepared by the study team. Data were extracted in two phases. In the first phase, citation details (author name, publication year, design, sample size, setting, population, intervention, comparison and outcomes) were extracted. In the second phase, the intervention results by group were extracted.

### Strategy for data synthesis

To obtain the pooled effects of the interventions, we conducted meta-analysis on the following outcomes for which there were an adequate number of studies with sufficiently 'similar' outcomes that could be pooled meaningfully: maternal stress, anxiety, depressive symptoms and birth weight (BW). We used the mean difference (MD) with the reported Standard Deviation (SD) of the outcome as a measure of effect size for each of the included studies. For the meta-analysis, the raw mean difference (D), with 95% CI across studies that measured the same outcome (depression with Edinburgh Postnatal Depression Scale (EPDS) or stress with Perceived Stress Scale (PSS), anxiety with State-Trait Anxiety Inventory (S-TAS) and birth weight in grams) was examined and presented. Sub-group analyses were performed to examine the existence of significant differences among studies that used different relaxation methods for any given outcome of interest. We assessed heterogeneity with the Cochrane's Q test and tau-squared ($T^2$) and measured inconsistency (the percentage of total variation across studies due to heterogeneity) of effects across relaxation interventions using the $I^2$ statistic. Publication bias was assessed using regression based on Egger's test. For all meta-analyses, random effects model using restricted maximum likelihood estimates (REML) were employed. Statistical significance was defined as $P < 0.05$. Stata version 16 software (College Station, Texas 77845 USA) was used for the meta-analyses and for visualizing the forest plots.

For outcomes where meta-analysis was not possible because of inadequate number of studies and small sample size, a narrative synthesis of the reviewed articles on the effect of the interventions on each outcome of interest was performed and reported. S2 Table shows the preferred reporting items for systematic review and meta-analysis we followed to report the findings.

## Results

### Search results: Final reviewed studies

A total of 32 studies were included in the systematic review. See Fig 1 for the flow diagram.

Four of the reviewed studies were quasi-experimental [32–35] and one was a non-randomized clinical trial [36]. The remaining 27 studies were RCTs. Among the 27 RCTs, 9 trials reported on maternal perceived stress during pregnancy using PSS [37–45], 13 trials reported on anxiety during pregnancy using the State-Trait Anxiety Inventory (S-TAI) [37, 38, 40, 42, 45–53], 7 trials reported on antenatal and postnatal depression using the Edinburgh Postnatal Depression Scale (EPDS) [32, 38, 44, 48, 49, 51, 54], and 8 trials reported on birth weight in grams or kilograms [33, 51, 55–60]. In addition, two trials reported the effects of antenatal relaxation on postnatal stress, anxiety and depression using the Depression, Anxiety and Stress Scale (DASS) [32, 34], three trials reported symptoms of maternal anxiety during labour and 24 hours postnatal using the Visual Analogue Scale for Anxiety (VAS-A) [54, 55, 61] and one trial reported anxiety using the pregnancy-related anxiety questionnaire [62]. Six trials reported on Apgar score [33, 51, 55–57, 60], three trials reported on gestational age (GA) [51, 58, 60], six trials reported on mode of delivery [33, 50–52, 55, 58], and four trials reported on duration of labour [50, 52, 55, 57].

### Study context/settings

Four of the studies were from a lower middle-income countries (India = 3, Egypt = 1), 14 from upper-middle-income countries (China = 1, Thailand = 1, Indonesia = 1, Turkey = 2, Malaysia = 3, Iran = 6), and 14 were from high-income countries (HIC; United States of America = 2, United Kingdom = 1, Germany = 1, Switzerland = 1, Greece = 1, Spain = 3, Taiwan = 5).

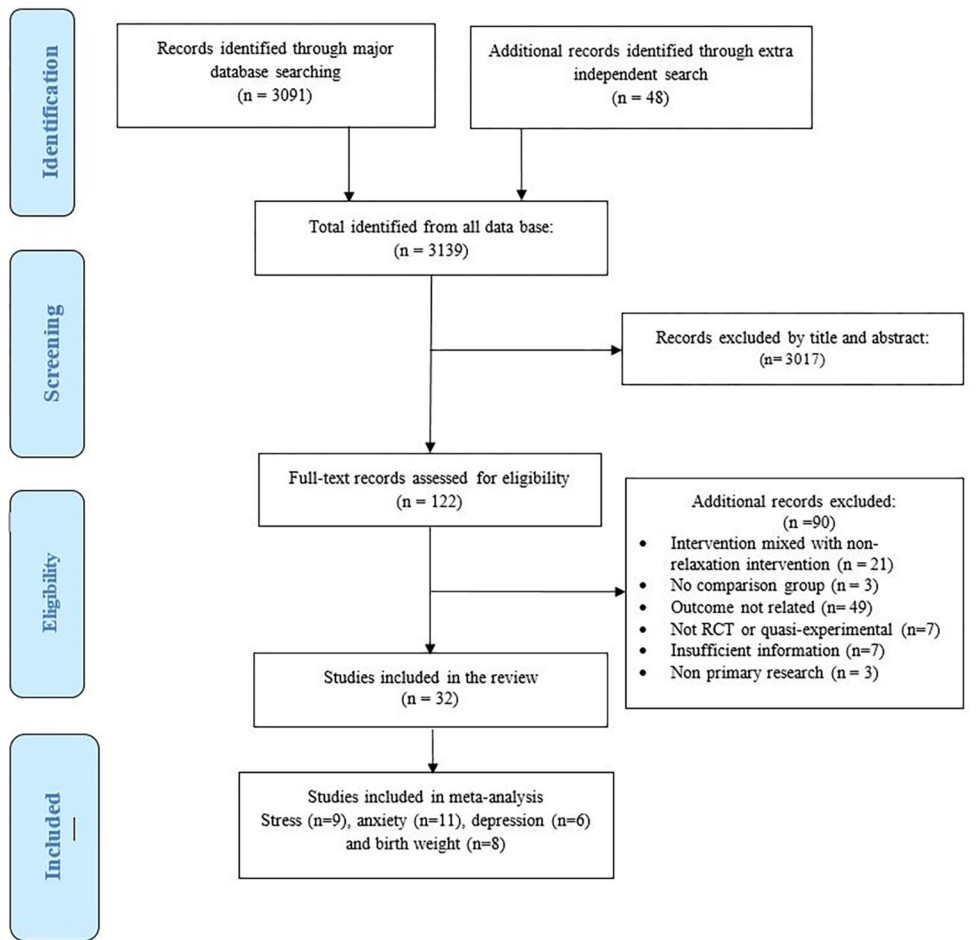

**Fig 1. PRISMA flow chart showing literature search results and study selection process.**

Trials examining outcomes of maternal stress, anxiety or depressive symptom were from USA = 2, UK = 1, Switzerland = 1, Greece = 1, Turkey = 2, China = 1, Spain = 2, India = 3, Egypt = 1, Indonesia = 1, Malaysia = 2, Iran = 4 and Taiwan = 5. Trials on birth outcomes were from India = 1, Turkey = 1, Thailand = 1, Malaysia = 1, Spain = 1 and Iran = 3. There were no published studies from sub-Saharan Africa or from other Low-Income Countries (LIC).

## Risk of bias within and across studies

Most studies had a low risk of selection, random allocation, concealment or other sources of bias. However, most studies had unclear risks on reporting bias (selective reporting of outcomes). S1 Table shows the risk of bias assessment findings for each of the studies.

## Characteristics of relaxation methods

The reviewed articles used one or a combination of the following relaxation methods: yoga = 5, music = 12, PMR/BR = 8, mindfulness = 4, hypnosis = 3, Benson relaxation with music = 1 and Benson relaxation alone = 1. The duration of interventions ranged from as short as a 10-minute brief experimental intervention to 6 months of daily relaxation practice. Table 1 provides a detailed description and summary of information on the included studies.

**Table 1. Description and summary of the results of the studies included in the systematic review and meta-analysis.**

| Authors, year | Study size | Country | Design | Relaxation, type and duration applied | Time outcome measured | Outcome | | Results Relaxation | Results Control |
|---|---|---|---|---|---|---|---|---|---|
| 1. Bastani F, et al. 2005 | EG:55 CG:55 | Iran | RCT | Progressive muscle relaxation (PMR) and breathing exercise for 7 weeks between gestational age of 14 and 28 weeks | Immeditely before and after the 7th week of intervention. Made group comparison | Stress—perceived stress scale (PSS) | PSS points | 24.4 ± 5.8 | 37.5 ± 5.7* |
| 2. Bastani F, et al. 2006 | EG:55 CG:55 | Iran | RCT | Progressive muscle relaxation (PMR) and breathing exercise for 7 weeks between gestational age of 14 and 28 weeks | Immeditely before and after intervention and at birth done for group comparisons | State anxiety—state anxiety trait inventory (S-STAI) | S-STAI points | 22.7 ± 7.4 | 38.5 ± 5.7‡ |
| | | | | | | Trait anxiety: (T-STAI) | T-STAI, points | 22.7± 7.4 | 38.5 ±5.7* |
| | | | | | | Birth weight (BW) | grams | 3168 ± 42 | 2883 ± 6* |
| | | | | | | Low BW | n (%) | 3 (5.8) | 14 (26.9)* |
| | | | | | | Gestational age | week | 38 ± 5.9 | 38 ± 4.40 ‡ |
| | | | | | | Preterm birth | n (%) | 1 (1.9) | 5 (9.8) ‡ |
| | | | | | | Mode of delivery; n (%) | Abnormal | 11 (21.2) | 25 (48.1)* |
| | | | | | | | SVD | 41 (78.8) | 27 (39.7)* |
| | | | | | | | C/S | 8 (15.4) | 21 (40.40)* |
| | | | | | | | Instrumental | 3 (5.8) | 4 (7.70)* |
| 3. Chuntharapat S, et al. 2008 | EG:33 CG:33 | Thailand. | RCT | Yoga 1 hour weekly for 6 Weeks at 26–28th, 30th, 32nd, 34th, 36th, and 37th week of gestation | After intervention (at birth) for group comparison | Birth weight | grams | 3076.8±311.2 | 3125.5±287.4‡ |
| | | | | | | Apgar score, 1st minute | ≤7; n (%) | 2 (6.1) | 5 (15.2) ‡ |
| | | | | | | | 8–10; n (%) | 31 (93.9) | 28 (84.8) ‡ |
| | | | | | | Apgar score, 5th minutes | ≤7; n (%) | 0 | 0 ‡ |
| | | | | | | | 8–10; n (%) | 33 (100) | 33 (100) ‡ |
| | | | | | | Length of labour, Minute | First stage | 520 ± 19 | 660 ± 27* |
| | | | | | | | Second stage | 27 ± 15 | 31 ± 14‡ |
| | | | | | | | Total labour | 559 ± 20 | 684 ± 28* |
| 4. Chang MY, et al. 2008 | EG: 116 CG: 120 | Taiwan | RCT | Music Therapy provided daily for 2 weeks | Pre/post difference for group comparison | Stress: PSS, points | Pretest | 17.4 ±4.6 | 16.7 ±4.3 |
| | | | | | | | Posttest | 15.3 ± 5.2 | 15.8 ± 6.0 |
| | | | | | | | Pre-post diff. | -2.1 | -0.9* |
| | | | | | | Anxiety: S-STAI, points | Pretest | 37.9 ±9.8 | 37.1±10.0 |
| | | | | | | | Posttest | 35.8 ± 10.9 | 37.8 ± 12.1 |
| | | | | | | | Pre-post diff | -2.1 | 0.7* |
| | | | | | | Depression -Edinburgh postnatal depression scale (EPDS), points | Pretest | 12.1±3.5 | 12.2±3.9 |
| | | | | | | | Posttest | 10.3 ± 4.1 | 12.1 ± 4.6 |
| | | | | | | | Pre-post diff. | 1.8 | 0.1* |
| 5. Satyapriya M, et al. 2009 | EG:45 CG:45 | India | RCT | Yoga daily in the 2nd and 3rd trimester | Pre/post difference for groups comparison | Stress: PSS points, group difference | 20th week of pregnancy | 15.9 ± 5.0 | 15.4±5.7‡ |
| | | | | | | | 36th week of pregnancy | 10.9 ±4.9 | 17.3±5.3* |
| | | | | | | | Pre-post diff | 5.0 | -1.9* |
| 6. Yang M, et al. 2009 | EG:60 CG:60 | China | RCT | Music therapy for 30 minutes on 3 consecutive days at admission for expected preterm birth | Pre/post difference for group comparison | Anxiety: STAI points, mean for pre/post and mean and SD for the pre/post difference reported | Pretest | 40.7 | 41.9‡ |
| | | | | | | | Posttest | 26.6 | 41.8* |
| | | | | | | | Pre-post diff: | -14.1±5.8 | -0.1±2.8* |
| 7. Urech C, et al. 2010 | EG1: 13 EG2: 13 CG:13 | Switzerland | RCT | Progressive Muscle relaxation and Guided imaginary experiment applied for 10 minutes only | Pre/post | State anxiety: S-STAI | | Groups did not differ significantly in change of state anxiety from pre to post intervention. Anxiety decreased equally in all three groups from pre-to post-relaxation,F(1,35) = 5.14, p = .030*, d = .38 | |
| 8. Liu YH, et al. 2010 | EG:30 CG:30 | Taiwan | RCT | Music therapy for 1 hour during labour | Posttest for group comparison | Labour anxiety using Visual Analogue Scale (VAS-A), points | Latent phase | 6.4 ± 3.0 | 5.2 ± 2.2‡ |
| | | | | | | | Active phase | 8.2 ± 2.3 | 7.7 ± 2.1‡ |
| | | | | | | | Latent and active phase diff. | -1.8 | 2.5* |

*(Continued)*

**Table 1.** (Continued)

| Authors, year | Study size | Country | Design | Relaxation, type and duration applied | Time outcome measured | Outcome | | Relaxation | Control (Results) |
|---|---|---|---|---|---|---|---|---|---|
| 9. Simavli S, et al. 2014 | EG:67 CG:65 | Turkey | RCT | Music therapy during labour | Pre/post for group comparison | Labour anxiety: VAS-A, points | Pretest | 2.8±0.4 | 2.7±0.4‡ |
| | | | | | | | Latent phase | 4.3 ± 0.8 | 5.1 ± 0.9* |
| | | | | | | | Active phase | 8.47 ± 0.7 | 9.4 ± 0.7* |
| | | | | | | | Second phase | 9.1 ± 0.6 | 9.8 ± 0.4* |
| | | | | | | | 2 h after delivery | 1.7± 0.3 | 4.2 ± 0.8* |
| | | | | | | Birth weight | G | 3375 ± 245 | 3420 ± 239‡ |
| | | | | | | Apgar 9/10 | n (%) | 67 (100%) | **61 (93.8%)** + |
| | | | | | | Duration of labour, Minutes | Latent phase | 162 ± 15 | 164 ± 15‡ |
| | | | | | | | Active phase | 189 ± 28 | 198 ± 15* |
| | | | | | | | Second phase | 83 ± 13 | 89 ± 18* |
| | | | | | | | Third stage | 17 ± 50 | 17 ± 5‡ |
| | | | | | | Mode of delivery: n (%) χ2 test, P>0.05 | Caesarean section | 5 (6.9) | 9 (12.2) ‡ |
| | | | | | | | Instrumental | 2(2.7) | 5 (6.8) ‡ |
| | | | | | | | Spontaneous vaginal delivery | 65 (90.2) | 60 (81.0) ‡ |
| | | | | | | | Episiotomy | 51 (76.1) | 52 (80.0) ‡ |
| | | | | | | Latent phase labour | SBP, mm Hg | 106.0±13.1 | 110.2±9.3* |
| | | | | | | | DBP, mmHg | 66.3±4.9 | 68.3±3.8* |
| | | | | | | | Hear rate | 76.0±4.8 | 78.7±5.9* |
| | | | | | | Active phase labour | SBP, mm Hg | 99.7±12.3 | 108.3±10.6* |
| | | | | | | | DBP, mmHg | 62.7±5.1 | 68.3±3.8* |
| | | | | | | | Hear rate | 74.4±4.9 | 78.7±5.8* |
| | | | | | | Second stage labour | SBP, mm Hg | 91.6±16.1 | 101.1±9.1* |
| | | | | | | | DBP, mmHg | 60.6±2.4 | 59.9±11.5‡ |
| | | | | | | | Hear rate | 73.9±3.8 | 76.5±3.7* |
| | | | | | | 2 h postpartum period | SBP, mm Hg | 94.2±5.0 | 99.9±15.5* |
| | | | | | | | DBP, mmHg | 59.4±2.4 | 63.4±10.7* |
| | | | | | | | Hear rate | 72.1±3.9 | 75.4±10.4* |
| 10. Simavli S, et al. 2014 | EG:71 CG:70 | Turkey | RCT | Music therapy during labour | Posttest group comparison | Antenatal depression: EPDS | Mean (SD) score | 8.0 (2.8) | 8.5 (2.6) ‡ |
| | | | | | | | EPDS≥10, n (%) | 18 (25.4) | 21 (30.0) ‡ |
| | | | | | | | EPDS≥13, n (%) | 8 (11.3) | 9 (12.9) ‡ |
| | | | | | | Postnatal depression day 1 | Mean (SD) score | 7.3±2.4 | 8.3±2.8* |
| | | | | | | | EPDS≥10, n (%) | 11 (15.5) | 22 (31.4)* |
| | | | | | | | EPDS≥13, n (%) | 4 (5.6) | 12 (17.1)* |
| | | | | | | Postnatal depression: EPDS, day 8 | EPDS, points | 7.1±2.1 | 8.6±2.9* |
| | | | | | | | EPDS≥10, n (%) | 9 (12.7) | 25 (35.7)* |
| | | | | | | | EPDS≥13, n (%) | 4 (5.6) | 13 (18.6)* |
| | | | | | | Postnatal Anxiety: VAS-A | VAS-A (1 h) | 3.3±0.5 | 4.9±0.9* |
| | | | | | | | VAS-A (4 h) | 2.7±0.4 | 4.2±0.8* |
| | | | | | | | VAS-A (8 h) | 2.3±0.3 | 3.3±0.5* |
| | | | | | | | VAS-A (16 h) | 1.7±0.3 | 2.8±0.4* |
| | | | | | | | VAS-A (24 h) | 0.9±0.6 | 2.3±0.3* |
| 11. Tragea C, et al. 2014 | EG:31 CG:29 | Greece | RCT | Breathing and progressive muscle relaxation 1–2 times a day for 6 weeks | Pre/post difference for group comparison | Stress: PSS, points | Pre-post diff | -3.7±1.8 | -0.5±1.8* |
| | | | | | | Anxiety: S-STAI | Pre-post diff | -3.5±2.8 | -2.0±2.9 ‡ |
| | | | | | | Anxiety: T-STAI | Pre-post diff | -3.8 (1.4) | -1.6 (2.5) ‡ |

(Continued)

**Table 1.** (Continued)

| Authors, year | Study size | Country | Design | Relaxation, type and duration applied | Time outcome measured | Outcome | | Relaxation | Results Control |
|---|---|---|---|---|---|---|---|---|---|
| 12. Guardino CM, et al. 2014 | EG:24 CG:23 | USA | RCT | Mindfulness training | Pre/post (immediate posttest and 6 weeks after post test) | PSS, points: Significant main effect of time: p < 0.05* | Pretest | 41.8±6.0 | 39.9±8.6‡ |
| | | | | | | | Posttest immediate | 37.3±5.4 | 35.8±8.0‡ |
| | | | | | | | Posttest 6 weeks | 36.2±5.9 | 37.4±7.3‡ |
| | | | | | | Anxiety: S-STAI: Significant main effect of time: p = 0.001* | Pretest | 45.7±7.6 | 44.4±11.0‡ |
| | | | | | | | Posttest immediate | 39.7±6.3 | 37.4±11.5‡ |
| | | | | | | | Posttest 6 weeks | 38.1±8.8 | 36.2±10.8‡ |
| 13. Newham J, et al. 2014 | EG:29 CG:22 | UK | RCT | Yoga training and practice applied for 8 weeks | Pre/post difference for group comparison | Anxiety: S-STAI, Points, medians (IQR) | Baseline | 28(24–42) | 32 (24–37) |
| | | | | | | | End line | 27(22–36) | 34 (25–38) ‡ |
| | | | | | | Anxiety: T-STAI, Points, medians (IQR) | Baseline: | 34 (29–40) | 35 (33–39) |
| | | | | | | | End line: | 34 (29–39) | 34 (30–41) ‡ |
| | | | | | | Depression: EPDS, Points, medians (IQR) | Baseline: | 5 (2–10) | 5 (4–8) |
| | | | | | | | End line: | 4 (2–7) | 6 (3–10)* |
| | | | | | | Anxiety: WDEQ | Baseline: | 74 (62–87) | 77 (60–85) |
| | | | | | | | End line: | 61 (42–77) | 69 (58–78)* |
| 14. Davis K, et al. 2015 | EG:23 CG:23 | USA | RCT | Yoga for 8 weeks | Pre/post, and midline assessment | Depression: EPDS, points | Baseline | 10.1 ±4.5 | 10.6±5.1 |
| | | | | | | | Midline | 8.5 ±4.9 | 8.8 ±6.0 |
| | | | | | | | End line | 6.4 ±4.0 | 7.3 ± 5.1 |
| | | | | | | | Baseline–end line mean diff. | 3.7 | 3.3‡ |
| | | | | | | Anxiety: S-STAI), points | Baseline | 36.9 ±12.2 | 41.7±10.8 |
| | | | | | | | Midline | 41.8±15.2 | 39.0±11.4 |
| | | | | | | | End line | 34.8±10.7 | 38.8 ±13.7 |
| | | | | | | | Baseline–end line mean diff. | 2.1 | 2.9‡ |
| | | | | | | Anxiety: T-STAI, points | Baseline | 45.0 ±12.1 | 45.4 ±10.2 |
| | | | | | | | Midline | 43.1 ±11.4 | 42.4±13.5 |
| | | | | | | | End line | 38.4 ±9.9 | 40.4±10.9 |
| | | | | | | | Baseline–end line mean diff. | 6.6 | 5‡ |
| 15. Chang HC, et al. 2015 | EG: 145 CG: 151 | Taiwan | RCT | Music therapy during 2nd and / or 3rd trimester | Pre/post for pre-post mean difference for group comparison | Stress: PSS, points | Pretest | 16.5±4.9 | 16.4±4.8 |
| | | | | | | | Posttest | 16.0 ±5.6 | 16.4 ±5.3 |
| | | | | | | | Pre-post diff. | 0.5 | 0‡ |
| | | | | | | STRESS: Pregnancy Stress Rating Scale (PSRS) | Pretest | 53.7±24.1 | 49.9±22.3 |
| | | | | | | | Posttest | 54.0±23.6 | 54.9±22.7 |
| | | | | | | | Pre-post diff. | 0.3 | 4.8* |
| 16. Liu YH, et al. 2016 | EG:61 CG:60 | Taiwan | RCT | Music therapy for 2 weeks | Pre/post | Stress: PSS, points | Pretest: | 17.1 ± 5.4 | 16.3 ± 5.2 |
| | | | | | | | Posttest: | 17.9 ± 4.1 | 19.3 ± 2.5 |
| | | | | | | | Pre-pots mean diff. for group comparison | -0.8 | -3.0* |
| | | | | | | Anxiety: S-STAI, points | Pretest | 39.7 ± 10.7 | 40.2 ± 10.2 |
| | | | | | | | Posttest | 37.3±10.0 | 42.1±11.6 |
| | | | | | | | Pre-pots mean diff. group comparison | 2.4 | -1.9* |

(*Continued*)

**Table 1.** (Continued)

| Authors, year | Study size | Country | Design | Relaxation, type and duration applied | Time outcome measured | Outcome | | Results Relaxation | Control |
|---|---|---|---|---|---|---|---|---|---|
| 17. Muthukrishnan S, et al. 2016 | EG:37 CG:37 | India | RCT | Mindfulness Meditation for 4 weeks from 13–16 gestational week | 5 weeks after enrollment (at 17–18 weeks of gestation) | Stress: PSS, points | Posttest between group comparison | 19.1 ±1.4 | 32.1±2.40* |
| | | | | | | BP: mmHg | SBP | 109.22±3.8 | 124.68±5.6* |
| | | | | | | | DBP | 69.11±2.23 | 69.11±2.2‡ |
| | | | | | | Hear rate variability | Beats/min | 26.59±2.1 | 20.65±1.5* |
| | | | | | | Respiratory rate | Breath/minute | 18.08 ±1.8 | 19.27±2.1* |
| | | | | | | Cold Pressor systolic blood pressure response. | | 9.68±1.8 | 13.38±2.23* |
| | | | | | | Cold Pressor diastolic blood pressure response. | | 4.19±0.98 | 7.54±1.4* |
| | | | | | | Mental arithmetic systolic blood pressure response. | | 8.97±2.21 | 13.49±3.1* |
| | | | | | | Mental arithmetic diastolic blood pressure response. | | 5.22±1.53 | 4.38±1.32 + |
| 18. Beevi Z, et al. 2016 | EG:28 CG:28 | Malaysia | Quasi-experimental | Hypnosis practiced since 16 week of gestation | Pre/post: | Stress: DASS-21, points at 36 weeks of gestation | Posttest mean group difference (raw data not given) | 5.8 ±5.4 | 10.7 ±8.9* |
| | | | | | | | | F (1,44) = 4.70, p = 0.03, partial $\eta^2$ = .101* | |
| | | | | | | Anxiety: DASS-21, points at 36 weeks of gestation | Posttest mean group difference (raw data not given) | F(1,44) = 10.76,p = 0.01, partial $\eta^2$ = 0.20* | |
| | | | | | | Depression: DASS-21, points at 36 weeks of gestation | Posttest mean group difference | F(1,16) = 0.958,p = 0.342, partial $\eta^2$ = .06. ‡ | |
| 19. Beevi Z, et al. 2017 | EG:23 CG:22 | Malaysia | Quasi-experimental | Hypnosis provided at 16, 20, 28, and 36 weeks of their pregnancy and advised to practice every day until labour | Posttest / done at birth | Birth weight, g | 3103.5±301.2 | | 3070.9±367.24‡ |
| | | | | | | SVD, n (%) | 19 (42.2) | | 14 (31.1) ‡ |
| | | | | | | C/S, n (%) | 4 (8.9) | | 8 (17.8) ‡ |
| | | | | | | Apgar score at 1 minute, % | 5 | 0 | 4.3‡ |
| | | | | | | | 6 | 0 | 4.3‡ |
| | | | | | | | 8 | 4.3 | 18.2‡ |
| | | | | | | | 9 | 95.7 | 72.7 * |
| | | | | | | Apgar score at 5 minute | 9 | 0 | 4.5‡ |
| | | | | | | | 10 | 100 | 95.5‡ |
| 20. Gonzalez et al. 2017 | EG: 204 CG: 205 | Spain | RCT | Music therapy for 40 minutes daily for 2 weeks | Posttest for group comparison | Birth weight | Kg | 3.4±0.4 | 3.4±0.5 ‡ |
| | | | | | | Newborn length | Cm | 50.3±1.86 | 50.6±2.0 ‡ |
| | | | | | | head circumference | Cm | 34.5±1.3 | 34.6±1.4 ‡ |
| | | | | | | Apgar score | 1 minute, | 9.1±0.8) | 9.0±0.9 ‡ |
| | | | | | | Apgar score | 5 minute, | 9.9±0.3 | 9.9±0.4 ‡ |
| 21. Novelia S, et al 2018 | EG:15 CG:15 | Indonesia | Quasi experimental | Yoga 2 times in 2 weeks each lasting 90 minutes | Posttest group comparison: | Anxiety (Anxiety: Hamilton Anxiety Rating Scale), n (%): (t = –9.83, p = 0.01)* | Yes | 2 (13.3%) | 15 (100%) |
| | | | | | | | No | 13 (86.7%) | 0 (0%) |
| 22. Gonzalez et al. 2018 | EG: 204 CG: 205 | Spain | RCT | Music therapy for 40 minutes daily for 2 weeks | Posttest group comparison | Onset of labour n (%),p < 0.01* | Spontaneous | 140 (68.63) | 111 (54.2) |
| | | | | | | | Stimulated | 5 (2.5) | 12 (5.9) |
| | | | | | | | Induced | 59 (28.9) | 82 (40.0) |
| | | | | | | Mode of delivery; n (%), P = 0.58‡ | Vaginal | 155 (75.9) | 151 (73.7) |
| | | | | | | | C/S, n (%) | 49 (24.02) | 54 (26.3) |
| | | | | | | Labour duration | First stage, hours | 4.36 ±3.7 | 5.54 ±4.8* |
| | | | | | | State-Trait-Anxiety (STA); posttest, group comparison | STA, points | 30.6±13.2 | 43.1 ±15.0* |
| 23. Beevi Z, et al. 2019 | EG:28 CG:28 | Malaysia | Quasi-experimental | Hypnosis provided at 16, 20, 28, and 36 weeks of their pregnancy and advised to practice every day until labour | 2 month postnatal mean (SD) difference for group comparison | Stress: DASS-21 | | | 3.6±5.1 ‡ |
| | | | | | | Anxiety: DASS-21 | | | 38.4± 58.8* |
| | | | | | | Depression: DASS-21 | 1.3± 2.4 | | 6.7± 5.7* |
| | | | | | | Depression: EPDS | 5.7±2.8 | | 10.6±4.0* |

(Continued)

**Table 1.** (Continued)

| Authors, year | Study size | Country | Design | Relaxation, type and duration applied | Time outcome measured | Outcome | | Relaxation | Control (Results) |
|---|---|---|---|---|---|---|---|---|---|
| 24. Pan Win Lan, et al. 2019 | EG:39 CG:35 | Taiwan | RCT | Mindfulness based Programs: Once every week for 8 weeks | Baseline and 3mo postpartum, mean (SD) | Stress: PSS, points | Pretest | 15.4(5.7) | **13.8(6.0)** |
| | | | | | | | Posttest | 11.6 ± 6.1 | **14.3 ±5.2** |
| | | | | | | | Pre-post diff | 3.8 | 0.5* |
| | | | | | | Depression: EPDS, points | Pretest | 9.5±4.0 | **8.7±4.5** |
| | | | | | | | Posttest | 6.5 ± 4.5 | **8.8 ±3.4** |
| | | | | | | | Pre-post diff | 3 | -0.1* |
| 25. Ahmadi M, et al. 2019 | EG:75 CG:75 | Iran | RCT | Progressive muscle relaxation/ PMR | Posttest between group comparison | Length at birth | CM | 52.1±3.6 | 48.6±3.4* |
| | | | | | | Birth weight | G | 3400±0.5 | 3200±0.6* |
| | | | | | | Postpartum depression, day 1 | Zung's Self-rating Depression Scale | 56.5±0.5 | 57.1±0.6‡ |
| | | | | | | Postpartum depression, day 3 | Zung's Self-rating Depression Scale | 49.7±0.4 | 59.4±0.7* |
| | | | | | | Postpartum depression, day 10 | Zung's Self-rating Depression Scale | 44±0.4 | 60.3±0.8* |
| 26. Rajeswari S, et al. 2020 | EG: 120 CG: 119 | India | RCT | Progressive Muscle Relaxation daily practice from 21/22 weeks of gestation until delivery | Posttest group comparison | Stress: Calvin Hobel scale: $P < 0.001^*$ | Minimal; n (%) | 0 (0.00) | 0 (0.0) |
| | | | | | | | Mild; n (%) | 51 (41.6) | 19 (15.2) |
| | | | | | | | Moderate; n (%) | 67 (54.4) | 71 (56.8) |
| | | | | | | | Severe; n (%) | 5 (4.00) | 35 (28.0) |
| | | | | | | | Overall stress; | 40.5 ±8.6 | 77.6 ±8.9* |
| | | | | | | Anxiety (S-STAI) Fisher exact test: F3 = 17.80, $P < 0.001^*$ | Minimal; n (%) | 0 (0.0) | 0 (0.0) |
| | | | | | | | Mild; n (%) | 22 (17.9) | 9 (7.2) |
| | | | | | | | Moderate; n (%) | 97 (78.9) | 84 (67.2) |
| | | | | | | | Severe; n (%) | 4(3.2) | 32 (25.6) |
| | | | | | | Anxiety (T-STAI) Fisher exact test: F3 = 18.60, $P < 0.001^*$ | Minimal; n (%) | 0 (0.00) | 0 (0.0) |
| | | | | | | | Mild; n (%) | 24 (10.0) | 10 (8.0) |
| | | | | | | | Moderate; n (%) | 95 (83.0) | 83 (66.4) |
| | | | | | | | Severe; n (%) | 4 (3.0) | 32 (25.6) |
| | | | | | | Overall anxiety: (STAI) Fisher exact test: F3 = 19.80, $P < 0.001^*$ | Minimal; n (%) | 0 (0.0) | 0 (0.0) |
| | | | | | | | Mild; n (%) | 26 (11.0) | 11 (8.8) |
| | | | | | | | Moderate; n (%) | 93 (82.0) | 82 (65.6) |
| | | | | | | | Severe; n (%) | 4 (32.0) | 32 (25.6) |
| | | | | | | Postpartum depression | EPDS, points | 6.9 ±2.5 | 10.5 ±2.7* |
| | | | | | | Gestational age, n (%) $P = 0.01^*$ | Before 37 weeks | 14 (11.5) | 25 (20.3) |
| | | | | | | | After 37 weeks | 108 (88.5) | 98 (79.7) |
| | | | | | | Gestational age (weeks) | | 38.0±3.6 | 37.2±4.2* |
| | | | | | | Apgar score; n (%); fisher exact test: P = 0.06‡ | 0–3 | 0 (0.0) | 3 (2.4) |
| | | | | | | | 4–6 | 2 (1.7) | 10 (8.2) |
| | | | | | | | 7–10 | 120 (98.3) | 110 (89.4) |
| | | | | | | Apgar score | Score/10 | 8.3 ±0.2 | 8.0 ±0.6‡ |
| | | | | | | Birth weight | Kg | 2.7 ±0.4 | 2.6 ±0.5* |
| | | | | | | Mode of delivery n (%): $P = 0.001^*$ | Normal vaginal | 90.0 (74.2) | 61.0 (49.6) |
| | | | | | | | Assisted vaginal | 5.0 (4.0) | 12.0 (9.8) |
| | | | | | | | C/S | 27 (21.8) | 50 (40.60) |
| | | | | | | Induced labour $P = 0.019^*$ | Yes, n (%) | 110(9.0) | 23 (20) |
| | | | | | | | No, n (%) | 111 (91.0) | 100 (80.0) |
| | | | | | | Hypertension $P = 0.037^*$ | Yes, n (%) | 4 (3.0) | 12 (10.0) |
| | | | | | | | No, n (%) | 118 (97) | 111 (90) |

(*Continued*)

**Table 1.** (Continued)

| Authors, year | Study size | Design | Country | Relaxation, type and duration applied | Time outcome measured | Outcome | Relaxation | Control |
|---|---|---|---|---|---|---|---|---|
| 27. Zarenejad M, et al. 2020 | EG:30 CG:30 | RCT | Iran | Mindfulness: 6 group counseling sessions twice a week, and each session lasted for 60 min | Posttest group comparison | Pregnancy-Related Anxiety: Posttest between group difference P = 0.001* — Pretest, | 182.9 ± 74.2 | 195.1 ± 42.9 |
| | | | | | | Posttest immediate | 154.5 ± 61.8 | 187.9 ± 41.5 |
| | | | | | | Posttest 1 month later | 124.9 ± 45.5 | 182.5 ± 41.7 |
| 28. Abd Elgwad FMH, et al. 2021 | EG:40 CG:40 | Non-randomized controlled clinical trial | Egypt | Benson's Relaxation twice daily (separated by 3 hours) for 3 days | Posttest group comparison | Stress: PSS, n (%) — Immediate posttest: P = 0.01* — Yes | 25 (62.5) | 40 (100) |
| | | | | | | No | 15 (37.5) | 0 (0) |
| | | | | | | Posttest after 3 days: P = 0.01* — Yes | 10 (25) | 40 (100) |
| | | | | | | No | 30 (75) | 0 (0) |
| | | | | | | Blood pressure: mmHg — SBP: Pretest | 158.3±12.2 | 150.5±18.8* |
| | | | | | | SBP immediate posttest | 144.1±12.1 | 145.1±17.1‡ |
| | | | | | | SBP 3 days posttest | 119.3±3.3 | 145.1±17.1* |
| | | | | | | DBP pretest | 95.2±11.2 | 90.8±12.2* |
| | | | | | | DBP immediate posttest | 87.8±9.5 | 87.4±9.8‡ |
| | | | | | | DBP 3 days posttest | 77.0±4.6 | 87.4±9.8* |
| | | | | | | Heart rate, Pretest | 92.3±7.2 | 97.8±16.3+ |
| | | | | | | Immediate posttest | 87.4±6.1 | 93.1±10.0* |
| | | | | | | 3 days posttest | 81.2±3.8 | 93.1±11.0* |
| | | | | | | Respiration rate — Pretest | 21.45±2.2 | 22.8±2.9* |
| | | | | | | Immediate posttest | 20.7±1.2 | 22.4±2.7* |
| | | | | | | 3 days posttest | 18.6±1.2 | 22.4±2.7* |
| 29. Bauer I, et al. 2021 | EG1:12 EG2:12 CG:12 | RCT (3-arm, parallel-group) | Germany | EG1: Music EG2: Guided imagery (GI) CG: Resting. Provided for 20 minute during labour | Pre/post | Indicators | Music group / GI group | Control group |
| | | | | | | Cardiovascular activity on heart rate, 5 minutes posttest | Music −2.33±2.9; GI −1.9±1.8 | −2.4±(3.4 |
| | | | | | | Cardiovascular activity on heart rate, 10 minutes posttest | Music −2.5±5.6; GI −1.4±3.0 | −2.6±4.3‡ |
| | | | | | | Heart rate variability | No significant group effect, $F_{(2,94)} = 0.624$, $p = .538$‡ | |
| | | | | | | Skin conductance 5 minute posttest | Music 0.01±0.1; GI 0.2± 0.7 | −0.1± 0.5‡ |
| | | | | | | Skin conductance 10 minute posttest | Music −0.04±0.2; GI −0.7±1.5 | −0.1±0.6‡ |
| 30. Estrella-Juarez F, et al. 2022 | EG1: 104 EG2: 124 CG: 115 | RCT 3-arm parallel group | Spain | EG1: Music therapy and EG2: Virtual reality (VR) intervention during labour | Pre/post | Outcomes | Music / VR | Control |
| | | | | | | First stage of labour, h | Music 4.7±4.1; VR 4.3±3.1 | 6.3±5.2* |
| | | | | | | Spontaneous labour n (%) | Music 50 (48.1); VR 103 (82.4) | 59±51.8* |
| | | | | | | Induction labour, n (%) | Music 54 (51.9); VR 22 (17.6) | 55 (48.2)* |
| | | | | | | SVD, n (%) | Music 49 (47.1); VR 73 (58.4) | 56 (49.1) ‡ |
| | | | | | | Instrumental assisted, n (%) | Music 21 (20.2); VR 32 (25.6) | 26 (22.8) ‡ |
| | | | | | | C/S, n (%) | Music 34 (32.7); VR 20 (16) | 32 (28.1) ‡ |
| | | | | | | Pretest T-STAI | Music 19.0± 6.9; VR 19.2±7.1 | 19.8±7.4 |
| | | | | | | Posttest T-STAI | Music 12.6±6.0; VR 12.4±5.9 | 19.2±9.0 |
| | | | | | | Pre-post mean diff. (within group comparison) | Music 6.4*; VR 6.8* | 0.6‡ |
| | | | | | | Pretest S-STAI | Music 16.3±5.8; VR 16.6±6.8 | 16.5±4.8 |
| | | | | | | Posttest S-STAI | Music 14.7±3.3; VR 15.2±3.3 | 17.6 ±7.2 |
| | | | | | | Pre-post diff (between group comparison) | Music 1.6‡; VR 1.3 ‡ | −1.1 ‡ |
| | | | | | | SBP | Music 106.9±8.3; VR 108.3±9.8 | 115.9±11.4* |
| | | | | | | DBP | Music 69.9±7.3; VR 70.4±7.9 | 75.0±8.9* |
| | | | | | | Heart rate | Music 79.2±8.4; VR 79.8±7.9 | 83.0±10.4* |

*(Continued)*

**Table 1.** (Continued)

| Authors, year | Study size | Country | Design | Relaxation, type and duration applied | Time outcome measured | Outcome | | Results | |
|---|---|---|---|---|---|---|---|---|---|
| | | | | | | | Relaxation | Control | |
| | | | | | | | | **CG** | |
| 31. Abarghoee SN, et al. 2022 | EG1: 35 EG2: 35 CG:35 | Iran | RCT (A parallel, three-armed) | Benson Relaxation Technique (BRT) and Music Therapy (MT) | Pre/post within and between group comparison | Anxiety: S-STAI; Pre-post difference within group comparison | MT group | CG | |
| | | | | | | | 49.4±1.6 | **50.3±1.4** | |
| | | | | | | | 43.1±1.2 | **48.3±1.7** | |
| | | | | | | | diff: 6.3* | diff: 2.0‡ | |
| | | | | | | | BRT group | | |
| | | | | | | | Pre: 50.6±1.3 | | |
| | | | | | | | Post: 42.3±1.3 | | |
| | | | | | | | diff: 8.3* | | |
| | | | | | | Anxiety (S-STAI) pre/post between group comparison | Pre: 50.6±1.3 | 50.3±1.4 ‡ | |
| | | | | | | | Post: 42.3±1.3 | 48.3±1.7* | |
| | | | | | | | 49.4±1.6 | | |
| | | | | | | | 43.1±1.2 | | |
| 32. Ghorbanneja d S, et al. 2022 | EG: 44 CG: 44 | Iran | RCT | Jacobson's progressive muscle relaxation | Posttest between group comparison | Birth weight | G | 2863.5±176.0 | 2762.7±202.1* |
| | | | | | | Birth length | CM | 47.8±2.1 | 47.5±2.2‡ |
| | | | | | | HC | CM | 34.7±0.2 | 34.5 0.1‡ |
| | | | | | | Gestational age | Week | 36.3±0.7 | 36.2±0.8‡ |
| | | | | | | Apgar score | 1st min | 9.0±0.4 | 8.8±0.3‡ |
| | | | | | | BP: mmHg | SBP | 137.6±3.9 | 147.5±5.0* |
| | | | | | | | DBP | 88.7±3.8 | 99.2±4.5* |
| | | | | | | **FBS** | **Mg/Dl** | **101.8±6.8** | 111.0±9.5* |

**Abbreviations:** ACTH, Adrenocorticotropic hormone; BP, Blood pressure; CG, CG, CM, Centimeter; Control group; C/S: Cesarean section; DASS-21, Depression, anxiety, stress scale- 21 items version; DBP, Diastolic blood pressure; EG, Experimental group; EPDS, Edinburgh postnatal depression scale; FBS, Fasting blood glucose; h, hour; HR, Heart rate; GI, Guided imaginary; IQR, Inter-quartile range; Kg, Kilogram; mg/dL, milligram per deciliter; Mo, Month; PSS, Perceived stress scale; Pre, pretest, post, posttest, diff, difference; RCT, Randomized control trial; SBP, Systolic blood pressure; SD, Standard deviation; S-STAI-S, State-trait anxiety inventory–state version; T-STAI, State-trait anxiety inventory–trait version; SVD, Spontaneous vaginal delivery; VAS, Visual analogue scale; WDEQ, Wijma delivery expectancy questionnaire–modified version.

Note

* P < 0.05

+ p = 0.05

‡ P ≥ 0.05

## Intervention outcomes

**Maternal mental health.** The effects of relaxation interventions on maternal mental health was examined in relation to symptoms of stress, anxiety or depression during the antenatal or postnatal periods.

*Maternal stress.* Nine trials (one of which reported stress at two time points) reported on the effectiveness of relaxation therapy on maternal stress symptoms using the PSS [37–45] and 2 trials using the DASS scale [32, 34]. Interventions applied were music therapy, meditation, mindfulness-based childbirth and parenting program, yoga, hypnosis, and PMR/BR.

A meta-analysis of the 9 trials (n = 1160 participants) using PSS mean and SD showed that relaxation interventions during pregnancy had a significant effect on reducing maternal perceived stress during pregnancy (overall mean difference (MD): -4.1; 95% CI: -7.4, -0.9)). In a subgroup analysis, only music therapy as a group significantly reduced maternal stress (MD: -.8, 95% CI: -1.53, -0.05), but not other relaxation methods. There was high level of heterogeneity among the studies (I$^2$ = 97.8%, P<0.01). Output of the meta-analysis on stress is provided in Fig 2.

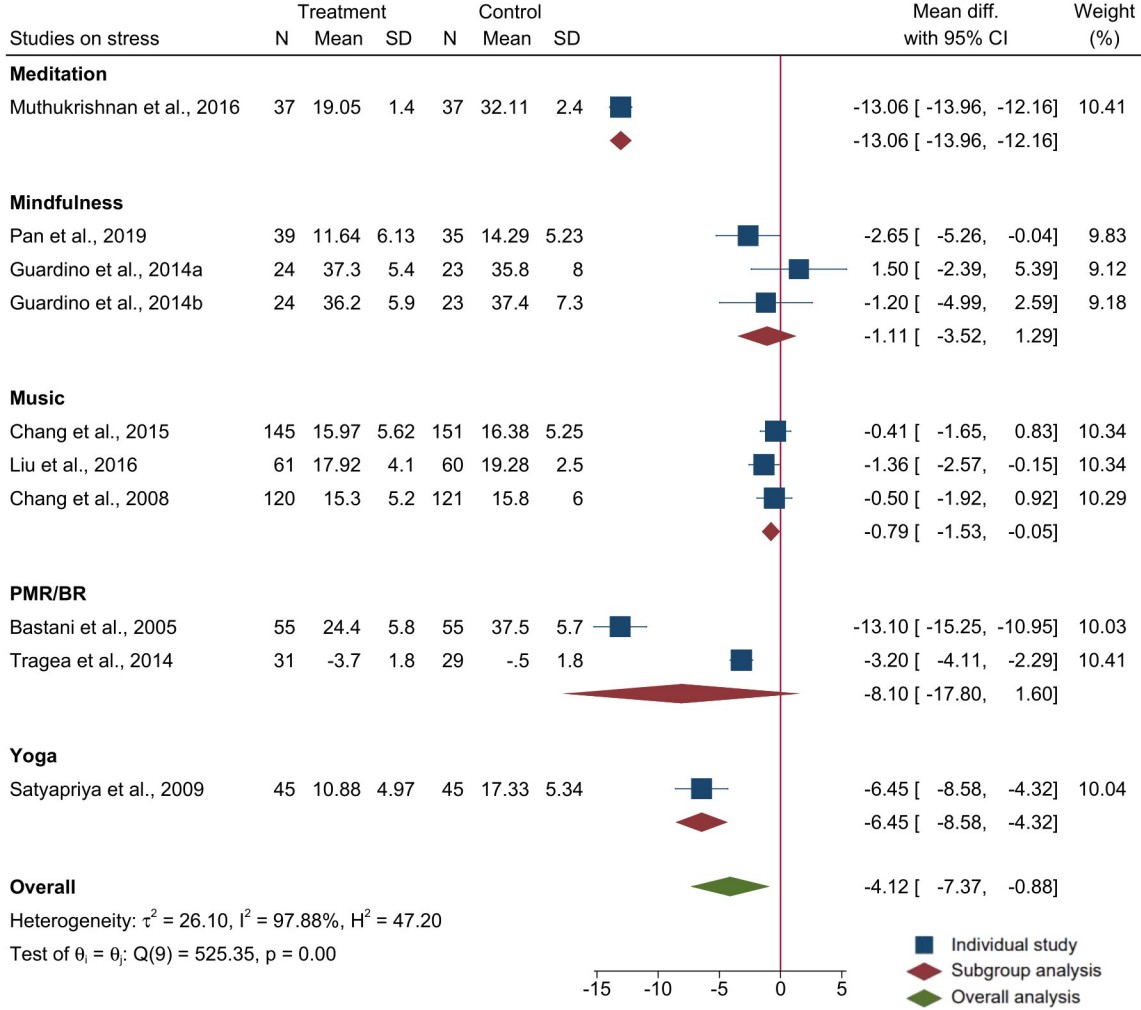

**Fig 2. Forest plot and subgroup analysis for raw mean difference of studies on the effects of relaxation interventions on maternal stress measured using the perceived stress scale (PSS).**

*Maternal anxiety.* Anxiety symptoms were reported in 13 trials [37, 38, 40, 42, 45–53] using the STAI, two trials using the DASS [32, 34], three using the VAS-A scale) [54, 55, 61] and one using pregnancy related anxiety questionnaire [62].

Eleven of the 13 trials reported mean and SD using S-STAI (two of which reported anxiety at two time points). Meta-analysis of the 11 trials (n = 1965 participates) showed that relaxation interventions provided during pregnancy were effective in reducing symptoms of maternal anxiety (overall MD: -5.04; 95% CI: -8.2, -1.9). In a subgroup analysis, only music therapy as a group showed a significant effect in reducing anxiety by 6 points (MD: -5.8; 95% CI: -9.1, -2.4), but not other relaxation methods. The trials were highly heterogeneous ($I^2$ = 97.1%, p<0.01). The output of the meta-analysis on anxiety is provided in Fig 3.

The other 3 trials that were not included in the meta-analysis (because they did not reported mean and SD) were also effective in reducing symptoms of anxiety during pregnancy [34], labour [55, 61], and during the 24 hours [54] and 2 months postnatal [32] periods.

*Maternal depressive symptoms.* Seven trials examined effects of relaxation interventions provided during pregnancy on maternal depressive symptoms measured using the EPDS [32,

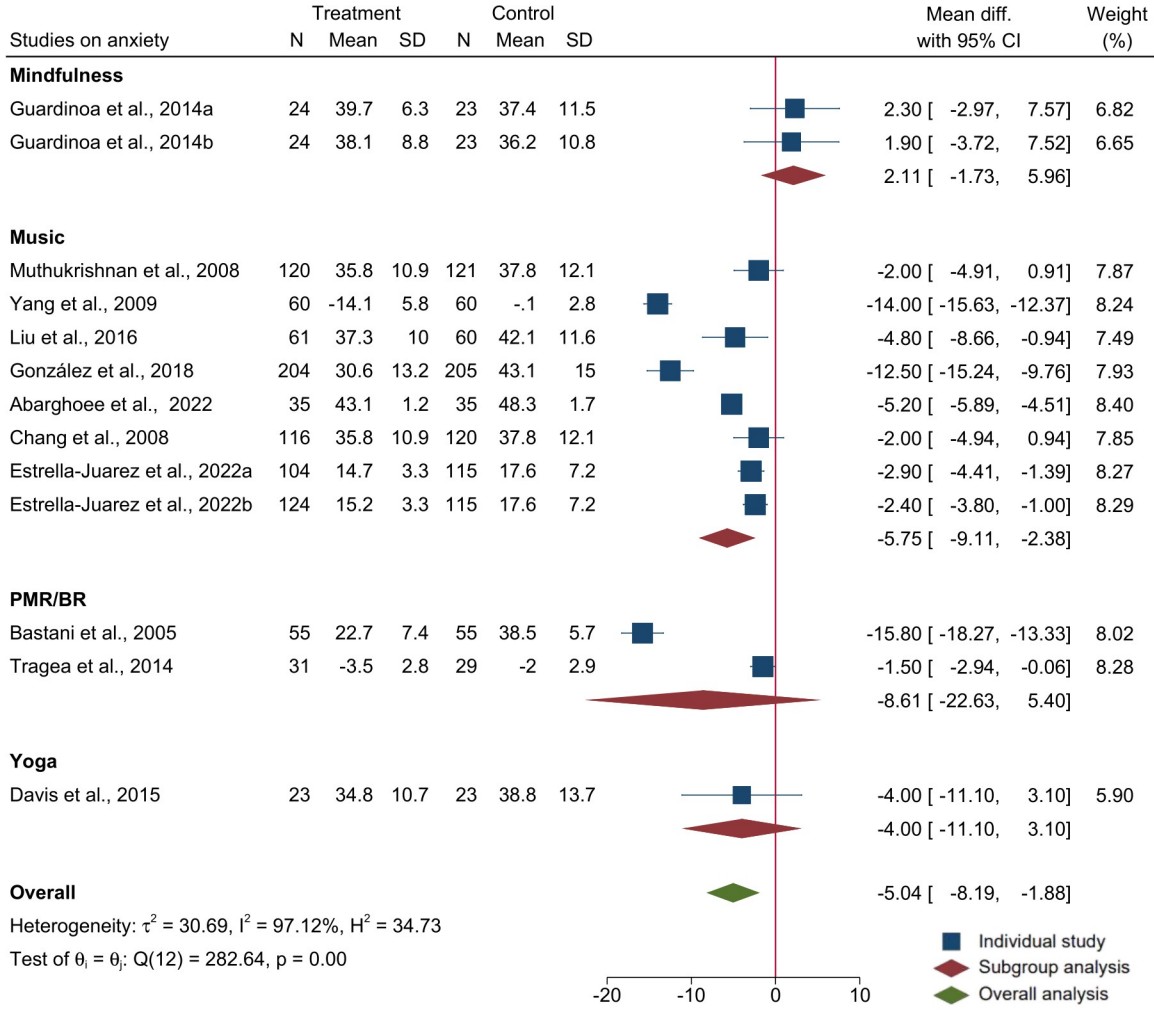

**Fig 3. Forest plot and subgroup analysis for raw mean difference of studies on the effects of relaxation interventions on antenatal anxiety score using S-TAI.**

38, 44, 48, 49, 51, 54]. Specific relaxation methods included in this section were yoga, Mindfulness-Based Childbirth and Parenting (MBCP) Music and PMR interventions.

Six of the 7 trials reported mean and SD using EPDS (one of which reported depression at two time points). Meta-analysis of the six trials (n = 933 participants) using EPDS mean and SD showed that relaxation interventions during pregnancy are effective in reducing maternal depressive symptoms (overall MD: -2.3; 95% CI: -3.4, -1.3) in the intervention compared to the control group. In a subgroup analysis, Music therapy as a group showed association with reduced depressive symptoms (MD: -2.2; 95% CI: -3.8, -0.06). The trials were found to be heterogeneous ($I^2$ = 83.4%, P<0.01). The effects of relaxation interventions in improving depression also persisted to immediate one week postnatal [55] and the two month postnatal [32] period. Output of the meta-analysis on depressive symptoms is provided in Fig 4.

**Newborn outcomes.** Birth weight (mean, SD) as an outcome was reported in 8 trials [33, 51, 55–60]. The meta-analysis of the 8 trials (n = 1239 participants) indicated that relaxation interventions improved birth weight (overall MD = 80; 95% CI: 1, 157). Subgroup analysis showed that only PMR/BR, but not other relaxation methods, increased birth weight significantly (MD = 165; 95% CI: 100, 231) in the intervention compared to the control group. The subgroup analysis showed significant heterogeneity among the studies ($I^2$ = 63.0%, P = 0.03). Output of the meta-analysis on birth weight is provided in Fig 5.

Apgar score as outcome was measured in 6 trials [33, 51, 55–57, 60]. A study in Turkey reported 100% of neonates born to mothers in the relaxation group (music therapy) compared to 93.8% of neonates born to mothers in the control group scored 9/10 (p = 0.05) for Apgar

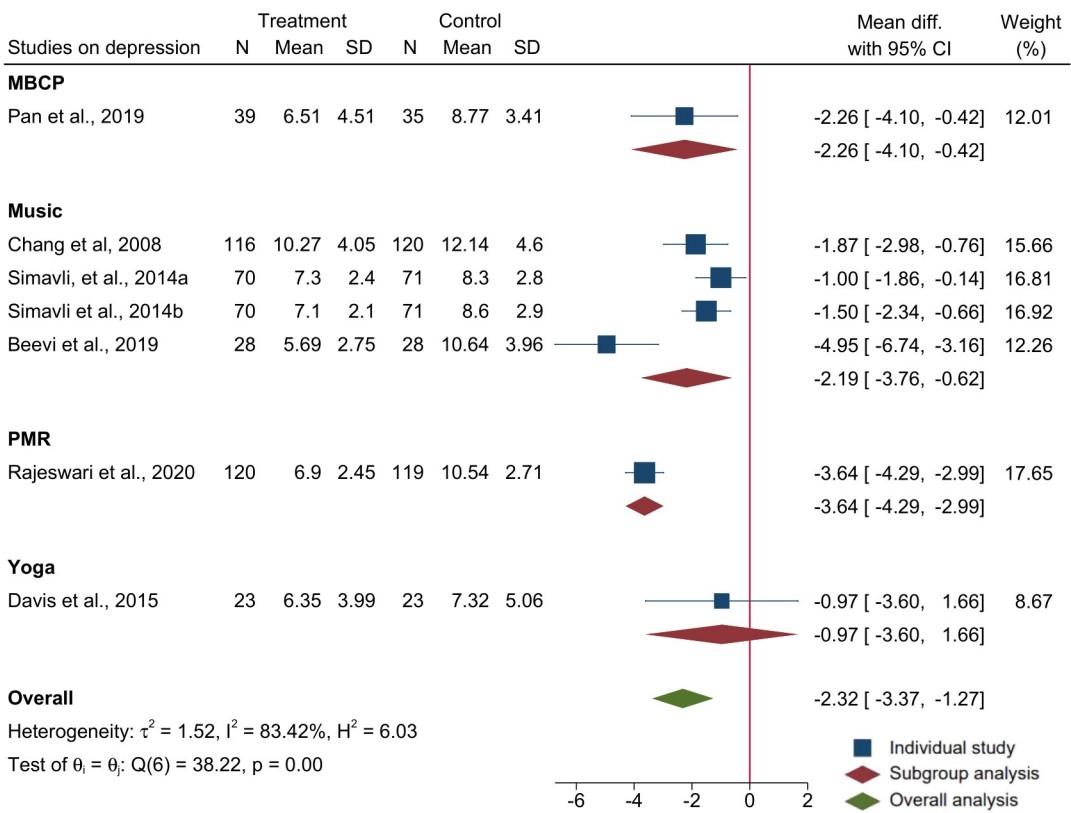

**Fig 4. Forest plot and subgroup analysis for raw mean difference of studies on the effects of relaxation interventions on depressive symptoms using EPDS.**

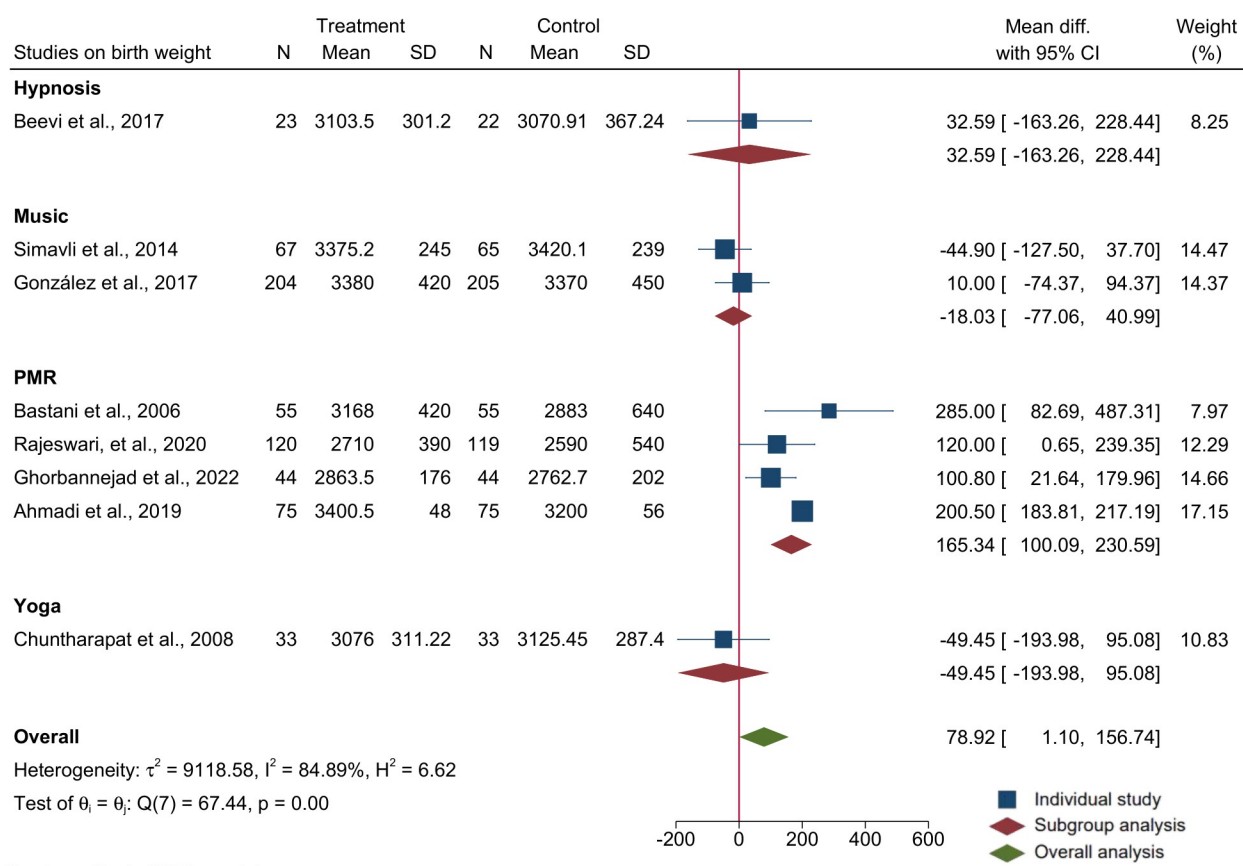

**Fig 5. Forest plot and subgroup analysis for raw mean difference of studies on the effects of relaxation interventions on birth weight (g).**

score at 5 minute evaluation [55]. The trial in Malaysia reported a significant difference in Apgar score at 1 minute evaluation where 96% of neonates born to mothers in the relaxation (hypnosis) group scored 9 compared to 73% of neonates born to mothers in the control group [33]. The other trials showed no effect on Apgar score either at 1 or 5 minute evaluation [51, 56, 57].

Gestational age as an outcome was reported in three trials [51, 58, 60]. In India, relaxation significantly increased gestational age at birth (38.0 (±3.6) weeks in the relaxation group vs 37.2 (±4.2) weeks in the control group, $p = 0.04$). The same trial reported that the percentage of preterm births was significantly lower in the relaxation group (11.5%) compared to the control group (20.3%) ($p = 0.01$) [51]. However, studies in Iran reported no significant effect of relaxation therapy on gestational age, but the sample size for this trial was small [58, 60]. Newborn length at birth was reported in three trials; one of which reported increased birth length in the relaxation group (mean difference 3.5 centimetres; 95% CI: 2.4, 4.6) [59]; but not the remaining trials [56, 60].

**Obstetric outcomes.** Four RCTs assessed the effect of relaxation interventions on obstetric outcomes. In Turkey, women who received music therapy during labour had a significantly shorter mean (SD) duration of labour 189 (28) minutes in the first stage of active labour and 83 (13) minutes in the second stage of labour compared to 198 (15) minutes and 89 (18) minutes respectively in the control group [55]. In the same trial, women in the intervention group (music therapy) had non-significantly decreased rates of Cesarean section (6.8%), instrumental delivery (2.7%), episiotomy (76.1%), and non-significantly increased rate of spontaneous

vaginal delivery (90.2%) compared to 12.2% cesarean section, 6.8% instrumental delivery, 81% episiotomy and 80% spontaneous vaginal delivery in the control group [55]. A study from Iran reported a significantly reduced rate of abnormal delivery in the relaxation (PMR/BR) group (21.2%) compared to 48.1%, p = 0.01, and an increased rate of spontaneous vaginal delivery (78.8%) compared to 39.7%, p = 0.01, in the control group [58]. Similarly, there was a significant improvement in spontaneous vaginal delivery (74.2% in the PMR/BR compared to 49.6% in the control group) and a decreased rate of Caesarian deliveries (21.8% in the PMR/BR compared to 40.6% in the control group) in India [51]. The same trial reported a significantly decreased rate of induced labour in the PMR/BR group compared to women in the control group ($F_2 = 5.50$, p = 0.019) [51].

An RCT in Thailand also reported significantly shorter duration of first stage labour 520 (186) minutes vs. 660 (273) minutes (p<0.05) and shorter total duration of labour 559 (203) minutes vs 684 (276) minutes (p<0.05) in the yoga group compared to women in the control group [57].

**Maternal physiological outcomes.** Measurements of maternal physiological outcomes were reported in six studies [36, 43, 52, 55, 60, 63]. Pregnant women in the relaxation group had lower systolic and diastolic blood pressure, heart rate, respiration rate and skin conductance level activity during pregnancy, labour and the postnatal period [36, 43, 52, 55, 60, 63].

## Discussion

In this systematic review and meta-analysis, we synthesized existing literature and provided up-to-date evidence on the effects of relaxation interventions during pregnancy on maternal mental health problems (stress, anxiety and depressive symptoms), and pregnancy and birth outcomes. Consistent beneficial impacts of relaxation interventions on mental health and birth weight outcomes were observed in terms of maternal stress, anxiety, depression and birth weight, although study heterogeneity was high. Furthermore, relaxation interventions consistently improved maternal physiological indicators during pregnancy and shortened the length of labour at birth. Findings on birth outcomes such as gestational age, mode of delivery, Apgar score and offspring birth length were mixed and non-conclusive. In subgroup analysis, music therapy has reduced symptoms of stress, anxiety and depression consistently and PMR/BR relaxation improved offspring birth weight. Several mechanisms such as brain stem reflex, arousal, inducing emotions, mental imagery, conjure episodic memory and evaluative conditioning, could be involved in music therapy to improve mental health [64–66]. On the other hand, PMR/BR, through activation of cortical brain activities and by enhancing blood circulation and oxygen saturation, could optimize mental health and physiologic output [67–70].

The meta-analysis indicated that relaxation interventions are effective in reducing stress during pregnancy. One underlying mechanism can be explained by the model of body-mind connection and integration [71] whereby body, mind, brain and behavior are all interlinked and influence one another [72, 73]. Psychological stress leads to sustained contraction of muscle tissues making them tense with increased vasoconstriction, blood pressure, heart rate and decreased circulatory outcomes until the stress is resolved. Physical relaxation methods, such as breathing and muscle relaxation, further contract and then relax the muscle to expel the newly induced stress along with the preexisting pathological stress from the body. Another mechanism is that psychological relaxations such as meditation and music therapies relax the mind, induce emotions, mental imagery, and counter unpleasant feelings and thoughts to improve mental wellbeing.

In addition to their effects on stress, the meta-analysis showed that relaxation interventions are effective in improving symptoms of anxiety and depression. This could be explained by the

fact that anxiety and depression are mainly the consequence of increased and unresolved stress in human life [74, 75]. Increased level of stress activates the HPA axis as well as the sympathetic and parasympathetic nervous system [8, 74, 75] and influences the neuronal circuits responsible for regulating and mediating anxiety and depression in the brain [8, 74, 75]. Thus, by reducing stress, relaxation therapy could break the neurobiological links between stress, anxiety and depression. Another mechanism of relaxation is through its effect on improving neurogenesis, synaptogenesis and increased gray matter density and volume with potential benefit for optimizing neurotransmitters in the brain [76, 77].

In two trials in this review, the positive effects of relaxation in improving anxiety and depression persisted into the postnatal period. This enduring benefit of relaxation could arise because relaxation interventions prevent antenatal anxiety and depression which would otherwise persist into the postnatal period. Alternatively, the relaxation interventions provided during pregnancy may have a prolonged effect on maternal stress management and reduce the risk of anxiety and depression in the postnatal period. Improved maternal well-being during pregnancy helps the mothers to care for herself more optimally during pregnancy while persistence of better maternal mental health into the postnatal period could help mother-infant attachment, child care and exclusive breastfeeding, all of which promote positive growth and development of the offspring [78, 79].

Finally, in the meta-analysis, relaxation interventions showed a positive effect on birth weight of the newborn. This was entirely explained by the effect of progressive muscle relaxation and deep breathing on birth weight [51, 58–60], whereas no effect was seen for music, hypnosis or yoga therapies [33, 55–57]. This contrast could be because deep breathing and muscle relaxation rather than music therapy improve physical relaxation and optimized maternal physiology to improve uterine circulation, benefitting fetal growth and development. However, PMR/BR interventions were also given for longer periods compared to yoga and music therapies which could result in stronger effects compared to the other approaches.

In the narrative synthesis, studies that evaluated physiological responses found relaxation therapies to be effective in improving the physiology of pregnant women by optimizing vital signs such as blood pressure, heart rate, body temperature and respiration. Inconsistent effects of relaxation interventions on pregnancy and birth outcomes such as mode of delivery, gestational age, birth length and Apgar score were observed. The lack of associations in some of the outcomes could be because most of the reviewed studies were primarily powered to examine impacts on maternal mental health but not obstetric and birth outcomes. In addition, only a few trials reported gestational age and birth outcomes, compounded by a relatively small sample size.

In summary, the mechanisms through which relaxation interventions could improve maternal well-being, and pregnancy and birth outcomes could involve an interplay of physical, psychological and physiological mechanisms. Physical responses to relaxation include immediate musculoskeletal relaxation and a decrease in muscle tension; psychological responses include mental calmness, silence and peace; and physiological responses to relaxation include optimized blood pressure, heart rate, respiratory rate and metabolic rate, along with decreased stress hormones and increased blood circulation [80–84]. Through one or a combination of these mechanisms, relaxation interventions could improve the health and well-being of pregnant woman, and this in turn may support fetal growth and development of the offspring.

## Strengths and limitations

A strength of this work is that we included trials that applied different forms of relaxation interventions and undertook both descriptive/narrative as well as pooled meta-analysis based

on data availability. However, the findings of the systematic review and meta-analysis also had some limitations. Most trials were primarily powered for maternal mental health, either stress, anxiety or depression, and not for pregnancy or birth outcomes. Because of lack of literature, some of the subgroup analysis involved only a single study. Furthermore, data on the effects of the relaxation interventions on neonatal outcomes other than birth weight were very limited and insufficient to conduct meta-analysis. Finally, most of the studies included in this review are from middle- or HIC and the findings might not be applicable for LIC settings, where both the sources of stress, and the feasibility of interventions, may be different.

## Conclusion and recommendation

The results of this review indicate that, in addition to physiological and mental health benefits for mothers, relaxation interventions improved birth weight and hold some promise for improving other newborn outcomes; therefore, this approach strongly merits further research. Future research that is adequately powered on birth and newborn outcomes such as gestational age, birth weight and birth length is crucial. Considering the magnitude of perinatal maternal mental health and psychological problems, the high burden of obstetric complications and the associated global burden of maternal and neonatal morbidity and mortality, the results of this review indicate that a range of complementary interventions may help address these problems. Their relative cost-effectiveness, ease and absence of adverse and teratogenic effects in comparison to pharmacological treatments favours the application of one or a combination of these relaxation therapies in this population group. Relaxation interventions are low-intensity and may be more scalable than individualized psychological interventions in resource-limited settings.

Therefore, we recommend that these relaxation interventions be evaluated for their acceptability, suitability and effectiveness to improve maternal and offspring health outcomes in LICs. Further evaluating the interventions in these settings would also be beneficial to understand whether, in places with severe food insecurity and a high burden of infections, which affect both maternal and infant health, relaxation interventions could mitigate the harmful effects of stressors.

## Supporting information

**S1 Table. Quality assessment report for risk of bias for included studies in the review based on the Cochrane Collaboration's risk of bias assessment tool.**
(DOCX)

**S2 Table. Preferred Reporting Items for Systematic reviews and Meta-Analyses extension for Scoping Reviews (PRISMA-ScR) checklist.**
(DOCX)

## Author Contributions

**Conceptualization:** Mubarek Abera, Charlotte Hanlon, Markos Tesfaye, Abdulhalik Workicho, Mary Fewtrell, Suzanne Filteau, Jonathan C. K. Wells.

**Data curation:** Mubarek Abera, Beniam Daniel, Abdulhalik Workicho.

**Formal analysis:** Mubarek Abera, Beniam Daniel, Abdulhalik Workicho.

**Methodology:** Mubarek Abera, Charlotte Hanlon, Beniam Daniel, Markos Tesfaye, Mary Fewtrell, Suzanne Filteau, Jonathan C. K. Wells.

**Project administration:** Mubarek Abera, Jonathan C. K. Wells.

**Supervision:** Charlotte Hanlon, Jonathan C. K. Wells.

**Visualization:** Mubarek Abera.

**Writing – original draft:** Mubarek Abera.

**Writing – review & editing:** Mubarek Abera, Charlotte Hanlon, Beniam Daniel, Markos Tesfaye, Abdulhalik Workicho, Tsinuel Girma, Rasmus Wibaek, Gregers S. Andersen, Mary Fewtrell, Suzanne Filteau, Jonathan C. K. Wells.

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
