## [Decision Letter · Decision Letter 0]

17 Jul 2023

PONE-D-22-31287Effect of relaxation interventions in pregnant women on maternal and neonatal outcomes: A systematic review and meta-analysisPLOS ONE

Dear Dr. Mubarak Abera 

Thank you for submitting your manuscript to PLOS ONE. After careful consideration, we feel that it has merit but does not fully meet PLOS ONE’s publication criteria as it currently stands. Therefore, we invite you to submit a revised version of the manuscript that addresses the points raised during the review process.

ACADEMIC EDITOR: Kindly check for errors in the manuscript 

"A total of 19 studies were included in the systematic review (Error! Reference source not found.)."====================

We look forward to receiving your revised manuscript.

Kind regards,

Fadhlun Alwy Al-beity, MD, PhD

Academic Editor

PLOS ONE

Journal Requirements:

3. Please ensure that you refer to Figures 1-4 in your text as, if accepted, production will need this reference to link the reader to the figure.

Reviewers' comments:

Reviewer's Responses to Questions

**Comments to the Author**

1. Is the manuscript technically sound, and do the data support the conclusions?

Reviewer #1: Yes

Reviewer #2: Yes

2. Has the statistical analysis been performed appropriately and rigorously? 

Reviewer #1: Yes

Reviewer #2: Yes

3. Have the authors made all data underlying the findings in their manuscript fully available?

Reviewer #1: Yes

Reviewer #2: No

4. Is the manuscript presented in an intelligible fashion and written in standard English?

Reviewer #1: Yes

Reviewer #2: Yes

5. Review Comments to the Author

Reviewer #1: 1. Pregnant women at risk of preterm births are likely to have more anxiety , depression etc. Why were they excluded from the analysis?

2. Although methodology section mentions to study impact in early postnatal life . Details are missing .

Reviewer #2: Dear authors

I would like to thank you for giving me the opportunity to review the manuscript entitled “Effect of relaxation interventions in pregnant women on maternal and neonatal outcomes: A systematic review and meta-analysis”. This systematic review and meta-analysis were conducted on studies that have tested relaxation interventions to improve maternal wellbeing, and pregnancy and birth outcomes in various settings. My comments are as follows:

- This study should be included in the review: Effects of Benson Relaxation Technique and Music Therapy on the Anxiety of Primiparous Women Prior to Cesarean Section: A Randomized Controlled Trial: https://www.hindawi.com/journals/arp/2022/9986587/

- Since it has been more than a year since the search, I suggest you do the search again.

- Some citation did not reveal in the text such as P 9.

6. PLOS authors have the option to publish the peer review history of their article (what does this mean?). If published, this will include your full peer review and any attached files.

Reviewer #1: **Yes: **DR PRAVEEN KUMAR

Reviewer #2: **Yes: **OK

---

## [Author Response · Author response to Decision Letter 0]

1 Sep 2023

Response to Feedback

Our general response: The authors are thankful to the reviewers for taking their time to read and give us important feedback/comments on our manuscript. Now we have considered the comments and have done important edits throughout the document to improve the quality and intensity of the manuscript. Below please find point-by-point response to the issues raised. We have used track change in the main document to indicate changes. 

1. Pregnant women at risk of preterm births are likely to have more anxiety, depression etc. Why were they excluded from the analysis?

Response: Thank you for this comment. Now we have included those articles in to the review. 

2. Although methodology section mentions to study impact in early postnatal life. Details are missing. 

Response: Now this is clarified that the aim of this study is to examine the effect of relaxation intervention on maternal wellbeing, pregnancy and birth and newborn outcomes. Our study does not assessed the impact during the early postnatal life. 

3. This study should be included in the review: Effects of Benson Relaxation Technique and Music Therapy on the Anxiety of Primiparous Women Prior to Cesarean Section: A Randomized Controlled Trial: https://www.hindawi.com/journals/arp/2022/9986587/

Response: Now we have added this and other articles identified when we update the searching. Thus a total of 13 articles are added now. Consequently we have revised the manuscript consistently throughout the text. 

4. Since it has been more than a year since the search, I suggest you do the search again. 

Response: Now we updated this on 26 August 2023 and made the desired revisions.

5. Some citation did not reveal in the text such as P 9.

Response: Now corrected

---

## [Decision Letter · Decision Letter 1]

3 Nov 2023

PONE-D-22-31287R1Effects of relaxation intervention during pregnancy on maternal mental health, and pregnancy and newborn outcomes: A systematic review and meta-analysisPLOS ONE

Dear Dr. Abera,

Thank you for submitting your manuscript to PLOS ONE. After careful consideration, we feel that it has merit but does not fully meet PLOS ONE’s publication criteria as it currently stands. Therefore, we invite you to submit a revised version of the manuscript that addresses the points raised during the review process.

We look forward to receiving your revised manuscript.

Kind regards,

Daniel Ahorsu, PhD

Academic Editor

PLOS ONE

Journal Requirements:

Additional Editor Comments:

The author must be commended for revising the manuscript according to the reviewers' comments. However, there are still minor revision that needs to be made.

Major comments

1. As authors keep on revising their manuscript, they must make sure the revision is reflected throughout the manuscript such that there are no anomalies in numbers (e.g., the number of studies for stress, and anxiety) and reports. For instance, findings reported in the results section were different from the Figures (e.g., Fig 1—PRISMA flow charts, Figure 2 etc). Also, for some of the Figures (Figure 2, 3), if the reference is the same, authors may differentiate them with “a” and “b” so readers may appropriately identify the exact study being referred to. For instance, Guardinoa et al. 2014a and Guardinoa et al. 2014b. If it is the same study, the same style can be used but it may be indicated beneath the figure as note “that the Guardinoa et al. 2014a and –2014b are from the same study. Please authors should make sure they rectify all these inconsistencies.

• Nine trials on maternal perceived stress during pregnancy using PSS (30–38), 12 trials on anxiety during pregnancy using the State-Trait Anxiety Inventory (S-TAI) (30,31,33,35,38–45), 7 trials on antenatal and postnatal depression using the Edinburgh Postnatal Depression Scale (EPDS) (31,37,41,42,44,46,47), and 6 trials reported birth weight in grams or kilograms (44,48–52)

2. The reviewer commented on a missing reference for a study that was added

In all, I suggest that the authors thoroughly go through and revise the manuscript to make sure there are no inconsistencies or missing data/information in the manuscript. Especially, they should go through the results section once more and update the discussion section appropriately.

Reviewers' comments:

Reviewer's Responses to Questions

**Comments to the Author**

1. If the authors have adequately addressed your comments raised in a previous round of review and you feel that this manuscript is now acceptable for publication, you may indicate that here to bypass the “Comments to the Author” section, enter your conflict of interest statement in the “Confidential to Editor” section, and submit your "Accept" recommendation.

Reviewer #1: All comments have been addressed

Reviewer #2: All comments have been addressed

2. Is the manuscript technically sound, and do the data support the conclusions?

Reviewer #1: Yes

Reviewer #2: Yes

3. Has the statistical analysis been performed appropriately and rigorously? 

Reviewer #1: Yes

Reviewer #2: Yes

4. Have the authors made all data underlying the findings in their manuscript fully available?

Reviewer #1: Yes

Reviewer #2: Yes

5. Is the manuscript presented in an intelligible fashion and written in standard English?

Reviewer #1: Yes

Reviewer #2: Yes

6. Review Comments to the Author

Reviewer #1: Thanks for making necessary changes . This study will help in management of mothers with psychological stress.

Reviewer #2: Dear authors

Thank you for addressing my comments.

Some studies that were added newly were not included in the reference list. For instance, Abarghoee SN, et al. 2022

7. PLOS authors have the option to publish the peer review history of their article (what does this mean?). If published, this will include your full peer review and any attached files.

Reviewer #1: **Yes: **DR PRAVEEN KUMAR

Reviewer #2: **Yes: **OK

---

## [Author Response · Author response to Decision Letter 1]

15 Nov 2023

Our general response: The authors are thankful to the reviewers for taking their time to read and give us important feedback/comments on our manuscript. Now we have considered the comments and have done important edits throughout the document to improve the quality and intensity of the manuscript. Below please find point-by-point response to the issues raised. We have used track change in the main document to indicate changes. 

1. As authors keep on revising their manuscript, they must make sure the revision is reflected throughout the manuscript such that there are no anomalies in numbers (e.g., the number of studies for stress, and anxiety) and reports. For instance, findings reported in the results section were different from the Figures (e.g., Fig 1—PRISMA flow charts, Figure 2 etc). Also, for some of the Figures (Figure 2, 3), if the reference is the same, authors may differentiate them with “a” and “b” so readers may appropriately identify the exact study being referred to. For instance, Guardinoa et al. 2014a and Guardinoa et al. 2014b. If it is the same study, the same style can be used but it may be indicated beneath the figure as note “that the Guardinoa et al. 2014a and –2014b are from the same study. Please authors should make sure they rectify all these inconsistencies.

Response: These now are corrected on the search flow diagram (fig 1), and all other figures (fig 2-5) by avoiding inconsistencies in the number and type of studies. Findings from similar studies are noted as ‘a’ and ‘b” as recommended by reviewer. 

2. Nine trials on maternal perceived stress during pregnancy using PSS (30–38), 12 trials on anxiety during pregnancy using the State-Trait Anxiety Inventory (S-TAI) (30,31,33,35,38–45), 7 trials on antenatal and postnatal depression using the Edinburgh Postnatal Depression Scale (EPDS) (31,37,41,42,44,46,47), and 6 trials reported birth weight in grams or kilograms (44,48–52)?

Response: this statement is now also revised and clarified in the main document as “Among the 27 RCT studies, 9 trials were on maternal perceived stress during pregnancy using PSS (35–43), 13 trials on anxiety during pregnancy using the State-Trait Anxiety Inventory (S-TAI) (35,36,38,40,43–51), 7 trials on antenatal and postnatal depression using the Edinburgh Postnatal Depression Scale (EPDS) (30,36,42,46,47,49,52), and 8 trials were on birth weight and reported weight in grams or kilograms (31,49,53–58).

3. The reviewer commented on a missing reference for a study that was added. In all, I suggest that the authors thoroughly go through and revise the manuscript to make sure there are no inconsistencies or missing data/information in the manuscript. Especially, they should go through the results section once more and update the discussion section appropriately.

Response: Now missing references are included (4 studies). Moreover the result and discussion section has been thoroughly edited. 

4. Some studies that were added newly were not included in the reference list. For instance, Abarghoee SN, et al. 2022 

Response: now this and other missed studies are included.

---

## [Editor Report · Decision Letter 2]

20 Nov 2023

PONE-D-22-31287R2Effects of relaxation interventions during pregnancy on maternal mental health, and pregnancy and newborn outcomes: A systematic review and meta-analysisPLOS ONE

Dear Dr. Abera,

Thank you for submitting your manuscript to PLOS ONE. After careful consideration, we feel that it has merit but does not fully meet PLOS ONE’s publication criteria as it currently stands. Therefore, we invite you to submit a revised version of the manuscript that addresses the points raised during the review process.

We look forward to receiving your revised manuscript.

Kind regards,

Daniel Ahorsu, PhD

Academic Editor

PLOS ONE

Journal Requirements:

Additional Editor Comments :

The authors have appropriately revised the manuscript but more needs to be done as mentioned in my previous comments.

Going through the data once more, I realized that the data were wrongly reported in Figure 2 (stress). Please recheck the mean difference for PMR/BR. It is not -13.1 but rather -8.1, (95%CI -17.80, 1.6) and this mean difference is not significant. Anytime a reader or researcher consuming your manuscript sees such an anomaly, it casts doubt on whether the whole manuscript can be trusted. This is a medical/health-related manuscript which will significantly impact how we treat people so data should be thoroughly checked before submission. Although reviewers are to help check these, the absolute responsibility rests with authors and not reviewers and editors.

The same issue, especially with PMR/BR can be found in anxiety figure 3. Again, although MD was -8.61 the 95%CI contains “0” which means it is not significant.

I am very sure I can get so many examples If I spend hours going through the manuscript again. So, I will entreat the authors to go through it once more or better still let another neutral person edit it before re-submission.

I also noticed that there were “Trails” instead of “trials” in the results (3 times).

Importantly, please prepare a response letter detailing point-by-point where the revision was made and with page numbers so I can quickly cross-check.

All the best.

---

## [Author Response · Author response to Decision Letter 2]

24 Nov 2023

Response to review feedback 

Our general response: we are really very thankful to the editor for giving us this opportunity to correct and make thorough edits to the manuscript. Accordingly all the authors including neutral reader have gone through the manuscript and made a detail read and added the necessary edits and corrections. Moreover below we have indicated for specific corrections we have made. 

1. Going through the data once more, I realized that the data were wrongly reported in Figure 2 (stress). Please recheck the mean difference for PMR/BR. It is not -13.1 but rather -8.1, (95%CI -17.80, 1.6) and this mean difference is not significant. 

Response: Again we thank you for this feedback. The confusion happen when we report finding from individual PMR/BR trial without reporting the finding from the overall subgroup analysis in the text. So now we have clarified and corrected this as follow on page 10. “In a subgroup analysis although PMR/BR as a group showed no significant association (mean difference -8.1, 95% CI: -17.8, 1.6), an individual PMR/BR trial (mean difference: -13.1; 95% CI:-15.3, -11.0) and meditation therapy (mean difference: -13.1; 95% CI:-14.0, -12.2) showed the highest effect size while music therapy as a group showed the lowest effect size (mean difference = -0.8; 95% CI:-1.5, -0.1).”

2. The same issue, especially with PMR/BR can be found in anxiety figure 3. Again, although MD was -8.61 the 95%CI contains “0” which means it is not significant.

Response: We have also corrected and clarified this as follows on page 11. “In the subgroup analysis although PMR/BR trials as a group showed no association (mean difference: -8.6; 95% CI: -22.6, 5.4), individual PMR/BR trials showed the highest effect size of -15.8 (mean difference: -15.8; 95% CI: -18.3, -13.3) and lowest effect size of -1.5 (mean difference: -1.5; 95% CI: -2.9, -0.1) in reducing symptoms of anxiety.”

3. Cortical spelling errors on Trial vs Trails: 

Response: We are very sorry for this and we have corrected them on three different sites in the main document page 9, 11 & 12. 

4. I will entreat the authors to go through it once more or better still let another neutral person edit it before re-submission.

Response: Thank you for this supportive feedback. Now the paper is thoroughly read and edited by all the authors and neutral reader and made important edits as can be seen in the track changed version of the main document.

---

## [Editor Report · Decision Letter 3]

30 Nov 2023

PONE-D-22-31287R3Effects of relaxation interventions during pregnancy on maternal mental health, and pregnancy and newborn outcomes: A systematic review and meta-analysisPLOS ONE

Dear Dr. Abera,

Thank you for submitting your manuscript to PLOS ONE. After careful consideration, we feel that it has merit but does not fully meet PLOS ONE’s publication criteria as it currently stands. Therefore, we invite you to submit a revised version of the manuscript that addresses the points raised during the review process.

We look forward to receiving your revised manuscript.

Kind regards,

Daniel Ahorsu, PhD

Academic Editor

PLOS ONE

Journal Requirements:

Additional Editor Comments :

The authors have revised the manuscript which has greatly improved the manuscript. However, I noticed that although the results have changed, it did not reflect in the discussion. Hence, I have provided a few comments to help the authors reflect on the results and discussion sections.

1. Authors should be consistent in how they report the sub-group analysis. A significant MD is what is important first and then the size of the MD and not the other way round. That is, if the MD is super high but it is not significant means nothing research-wise although noteworthy. Hence, I may suggest that authors report those MDs that are significant and then the MD sizes of those significant MDs. Then other super high non-significant MD may be mentioned as noteworthy cases.

2. Authors may choose to mention that the MD of for instance PMR/BR is the highest but the point is that it is not significant and so, research-wise, it does not matter as it can be equated to the other non-significant sub-group analysis. The main point is to clearly discuss why you had these results in the discussion section. In other words, clearly discuss the reasons behind the high MD but non-significant results. This applies to stress and anxiety.

3. Again, if authors will be mentioning sub-group results, then they should be ready to discuss those results. The discussion should also mention the limitations including the effect of single studies for some sub-groups. This is to help consumers understand the results clearly and know the pros and cons of using your findings. As mentioned previously, this manuscript will go a long way to influence clinical decisions and so sentences should be constructed with great caution.

4. Limitations: single studies in some subgroups analysis and their effect

---

## [Author Response · Author response to Decision Letter 3]

15 Dec 2023

Our response: Again we are very thankful to the editor for giving us these review feedback. Accordingly we have revised and made a careful necessary corrections. Below we have indicated for specific corrections we have made. 

1. Authors should be consistent in how they report the sub-group analysis. A significant MD is what is important first and then the size of the MD and not the other way round. That is, if the MD is super high but it is not significant means nothing research-wise although noteworthy. Hence, I may suggest that authors report those MDs that are significant and then the MD sizes of those significant MDs. Then other super high non-significant MD may be mentioned as noteworthy cases.

Response: Thank you for this comment. Now we have corrected this accordingly and followed a consistent reporting for subgroup analysis in the text. Corrections are made on page 10-12. 

2. Authors may choose to mention that the MD of for instance PMR/BR is the highest but the point is that it is not significant and so, research-wise, it does not matter as it can be equated to the other non-significant sub-group analysis. The main point is to clearly discuss why you had these results in the discussion section. In other words, clearly discuss the reasons behind the high MD but non-significant results. This applies to stress and anxiety.

Response: Yes we agree and accept this critical comment. Now we have chosen to report in the text only those that are statistically significant and discussed them accordingly. Corrections are made on page 10-12 and also page 14-16. 

3. Again, if authors will be mentioning sub-group results, then they should be ready to discuss those results. The discussion should also mention the limitations including the effect of single studies for some sub-groups. This is to help consumers understand the results clearly and know the pros and cons of using your findings. As mentioned previously, this manuscript will go a long way to influence clinical decisions and so sentences should be constructed with great caution.

Response: This is also well addressed in the discussion and limitation sections. Corrections are made on page 14-16. 

4. Limitations: single studies in some subgroups analysis and their effect

Response: This is now added in the limitation section as “Because of lack of literature, some of the subgroup analysis involved only single study”. Corrections are made on page 17.

---

## [Editor Report · Decision Letter 4]

26 Dec 2023

Effects of relaxation interventions during pregnancy on maternal mental health, and pregnancy and newborn outcomes: A systematic review and meta-analysis

PONE-D-22-31287R4

Dear Dr. Abera,

We’re pleased to inform you that your manuscript has been judged scientifically suitable for publication and will be formally accepted for publication once it meets all outstanding technical requirements.

Kind regards,

Daniel Ahorsu, PhD

Academic Editor

PLOS ONE
---

## [Editor Report · Acceptance letter]

17 Jan 2024

PONE-D-22-31287R4 

PLOS ONE

Dear Dr. Abera, 

I'm pleased to inform you that your manuscript has been deemed suitable for publication in PLOS ONE. Congratulations! Your manuscript is now being handed over to our production team.

Kind regards, 

on behalf of

Dr. Daniel Ahorsu 

Academic Editor

PLOS ONE